# Deficiency of neuronal LGR4 increases energy expenditure and inhibits food intake via hypothalamic leptin signaling

Liping Zhang [1,2], Yuan Li[1,2], Wenbin Gao [1], Ziru Li[1], Tong Wu [1], Chunhui Lang[1], Liangyou Rui [1✉] & Weizhen Zhang [1✉]

## Abstract

The metabolic effects of leucine-rich repeat-containing G protein-coupled receptor 4 (LGR4) remain largely unknown. Here, we showed that knockdown of *Lgr4* in nestin progenitor or Sp1 mature neurons reduced high fat diet (HFD)-induced obesity by increasing energy expenditure and inhibiting food intake. Deficiency of LGR4 in AgRP neurons increased energy expenditure, and inhibited food intake, leading to alterations in glucose and lipid metabolism. Knock-down of *Lgr4* in Sf1 neurons enhanced energy expenditure, reduced adiposity, and improved glucose and lipid metabolism. The metabolic benefits of neuronal LGR4 occurred via improvement of leptin signaling in AgRP and Sf1 neurons. Knockdown of *Lgr4* in nestin, Sp1, AgRP or Sf1 neurons decreased hypothalamic levels of SOCS-3, and increased phosphorylation of STAT3. These alterations were associated with a significant reduction in the hypothalamic levels of β-catenin. Inhibition of β-catenin signaling by Dkk1 significantly attenuated the decrement of phospho-STAT3 and concurrent increase of SOCS-3 induced by Rspondin 3, an endogenous ligand for LGR4. Our results thus demonstrate that hypothalamic LGR4 may promote energy conversation by increasing food intake and decreasing energy expenditure. Deficiency of neuronal LGR4 improves hypothalamic leptin sensitivity via suppression of β-catenin signaling.

**Keywords** Obesity; Leucine-rich Repeat-containing G Protein-coupled Receptor 4 (LGR4); Hypothalamus; β-catenin; Leptin Signaling
**Subject Categories** Metabolism; Neuroscience

## Introduction

Obesity, now a worldwide health problem, is a major risk factor for type 2 diabetes, non-alcoholic fatty liver diseases, cardiovascular disease, and some types of cancer (Bhupathiraju and Hu, 2016). Obesity is a consequence of long-term energy surplus resulting from disruption of the balance between energy intake and expenditure (Hill et al, 2012). Under physiological conditions, energy balance is controlled by the hypothalamus (Waterson and Horvath, 2015). A variety of chemical messages from neuronal and peripheral tissues activates receptors, many G protein coupled receptors (GPCR), within the hypothalamus to orchestrate energy homeostasis. For example, the melanocortin 4 receptor (MC4R) in the hypothalamus modulates sympathetic outflow to increase energy expenditure (Yong et al, 2021). For the adrenergic receptors, α1 and α2 subtypes are abundantly expressed in the hypothalamus (Domingos-Souza et al, 2021). Activation of α1 receptors depolarizes hypothalamic neurons via Gαq-mediated increase of intracellular calcium and closure of G protein-coupled inwardly rectifying potassium channels (Hein, 2006). In contrast, α2 activation hyperpolarizes hypothalamic neurons via Gαi-mediated suppression of voltage-gated calcium channels and opening of G protein-coupled inwardly rectifying potassium channels (Hein, 2006). In this way, the sympathetic tone of the hypothalamus is determined by the interplay of α1 and α2 adrenergic receptor in distinct neurons. Chemical signals from peripheral tissues also function through GPCRs in the hypothalamus to control food intake and energy expenditure (Lenard and Berthoud, 2008). For example, hypothalamic glucagon-like peptide (GLP-1) receptor has been demonstrated to mediate the anorexigenic effects of GLP-1 and GLP-1 analogues such as liraglutide and semaglutide (O'Neil et al, 2018) (Pi-Sunyer et al, 2015). Ghrelin, a gastric hormone secreted by X/A like endocrine cells, activates its receptor growth hormone secretagogue receptor 1a (GHSR1a) in hypothalamic neurons to stimulate food intake (Cowley et al, 2003). These observations indicate that targeting hypothalamic GPCRs may hold promise as interventions for obesity and its related metabolic dysfunction.

Leucine-rich repeat-containing G protein-coupled receptors (LGRs), members of the rhodopsin GPCR family identified by the unique nature of their long leucine-rich repeat extracellular domains, include three subtypes: LGR4, 5 and 6 (Barker et al, 2013). Being mainly expressed in the stem cells, LGRs are critical for the maintenance of stem cells and for embryonic development (Li et al, 2015; Luo et al, 2013). Unlike LGR5 and LGR6 subtypes, LGR4 is expressed not only in stem cells but also in differentiated cells such as adult enterocytes, adipocytes, hepatocytes and hypothalamic neurons (Li et al, 2015; Yi et al, 2013). The presence of LGR4 in these cells

[1]Department of Surgery, University of Michigan Medical Center, Ann Arbor, MI 48109, USA. [2]These authors contributed equally: Liping Zhang, Yuan Li.
✉E-mail: ruily@umich.edu; weizhenz@med.umich.edu

suggest its relevance to metabolism control. This concept is supported by recent studies. Mice with global ablation of LGR4 demonstrate a decrement in adiposity and resistance to dietary and leptin mutant-induced obesity (Wang et al, 2013). This alteration is associated with an increase in the browning of white adipose tissues and energy expenditure (Wang et al, 2013). Human genetic analysis has identified the gain-of-function variant of LGR4, LGR4 A750T variant, as a genetic determinant of central obesity (Zou et al, 2017). However, it is unclear whether the metabolic benefits of LGR4 deficiency are attributed to its peripheral action on adipocytes and hepatocytes or to central actions. Suggesting central effects, activation of LGR4 by R-spondin 1 or 3 in hypothalamus has been demonstrated to inhibit food intake (Li et al, 2014). This observation suggests that the metabolic action of neuronal R-spondin-LGR4 signaling may be distinct.

In this study, we examined the role of hypothalamic LGR4 signaling in energy homeostasis. Its metabolic effects on body weight, body mass composition, food intake, energy expenditure, and the underlying neuronal circuits involved were evaluated using a series of transgenic mouse models with deficiency of LGR4 in specific neurons. In addition, we explored the interaction of hypothalamic LGR4 signaling with leptin signaling and the sympathetic nervous system (SNS) axis. Deficiency of hypothalamic LGR4 rendered mice resistant to diet-induced-obesity via increasing energy expenditure and reducing food intake. Hypothalamic LGR4 deficiency in agouti-related protein (AgRP) neurons of the arcuate nucleus (ARC) and Sf1 neurons of the ventromedial hypothalamus (VMH) increased leptin sensitivity and SNS response by reducing the levels of suppressor of cytokine signaling 3 (SOCS-3), an inhibitor of leptin signaling. This likely occurred through suppression of β-catenin. Our results thus demonstrate that deficiency of hypothalamic LGR4 affects whole body metabolism by activating the leptin signaling-SNS axis.

# Results

## Knockdown of Lgr4 in *nestin* neurons renders mice resistant to high fat diet-induced obesity by increasing energy expenditure and inhibiting food intake

We have previously demonstrated the expression of *Lgr4* mRNA in rodent brain, specifically in the hypothalamus, cortex, hippocampus and amygdala. In the hypothalamus, *Lgr4* is abundant in the ventromedial nucleus (VMH), arcuate nucleus (ARC), median eminence (ME) and in ependymocytes lining the third ventricle (Li et al, 2014). To investigate the physiological functions of neuronal LGR4, we generated neuron-specific LGR4 knockout mice (*Nestin-Lgr4⁻/⁻*) by crossing *Lgr4*^flox/flox^ (FF) mice with *Nestin-Cre* mice as shown in Appendix Fig. S1A. Western blotting results indicated a substantial deficiency of LGR4 in the whole brain (Fig. 1A). Since no significant difference in body weight, fat mass and lean mass was observed between *Nestin-Cre* and FF mice fed either with NCD or HFD (Appendix Fig. S1B,C), we used littermate FF mice as controls in the following experiments. Relevant to FF mice, *Nestin-Lgr4⁻/⁻* mice fed NCD showed no significant difference in body weight (Appendix Fig. S1D), body composition, food intake or tissues weights (Appendix Fig. S1E–G), glucose tolerance, insulin tolerance, plasma levels of insulin and leptin (Appendix Fig. S1H–K). In mice fed 60% HFD for

16 weeks, deficiency of LGR4 in nestin neurons demonstrated a significant resistance to HFD induced obesity relative to the littermate FF mice (Figs. 1B and EV1A). Whole body fat mass was substantially decreased in *Nestin-Lgr4⁻/⁻* mice fed HFD, while lean mass remained unchanged (Figs. 1C and EV1B). Tissue weights of liver, subcutaneous adipose tissue (SAT), and brown adipose tissue (BAT) were significantly reduced in *Nestin-Lgr4⁻/⁻* mice relative to FF mice. Gonadal adipose tissue (GAT) weight of female *Nestin-Lgr4⁻/⁻* mice was lower than sex matched FF mice (Figs. 1D and EV1C). These results demonstrate that LGR4 deficiency in nestin neurons renders mice resistant to diet-induced obesity.

Body weight and adiposity are controlled by the balance between food intake and energy expenditure. Thus, we examined food intake and energy expenditure in the *Nestin-Lgr4⁻/⁻* mice. Consistent with the lean phenotype, both cumulative food intake and weekly food intake from male and female *Nestin-Lgr4⁻/⁻* mice were significantly reduced relevant to sex and age matched FF mice (Figs. 1E and EV1D). Further, $O_2$ consumption and $CO_2$ production in male and female *Nestin-Lgr4⁻/⁻* mice were significantly higher relative to sex and age matched FF mice both in dark and light phases (Figs. 1F,G and EV1E,F; Appendix Fig. S6A,B). Respiratory exchange ratio (RER) was lower in *Nestin-Lgr4⁻/⁻* mice than FF mice (Figs. 1H and EV1G). In addition, activity and heat production in male and female *Nestin-Lgr4⁻/⁻* mice were higher relative to sex and age matched FF mice (Figs. 1I,J and EV1H,I; Appendix Fig. S6C). Both male and female *Nestin-Lgr4⁻/⁻* mice showed higher body temperature under a short cold exposure at 4 °C relative to sex and age matched FF mice (Figs. 1K and EV1J). These results demonstrate that deficiency of LGR4 in neurons protects mice from obesity by reducing food intake and increasing energy expenditure.

Next, we analyzed whether neuronal LGR4 deficiency can ameliorate HFD-induced dysfunction of glucose and lipid metabolism. Both male and female *Nestin-Lgr4⁻/⁻* mice showed improved glucose tolerance (Figs. 1L and EV1K) and insulin sensitivity relative to FF mice (Figs. 1M and EV1L). Basal plasma insulin (Figs. 1O and EV1N) as well as glucose-stimulated insulin secretion after 16 h fasting (Figs. 1N and EV1M) in *Nestin-Lgr4⁻/⁻* mice were reduced relative to sex and age matched FF mice. Lipid droplets in hepatocytes of *Nestin-Lgr4⁻/⁻* mice were smaller and less abundant relative to FF mice (Fig. 1P). Levels of triglyceride and cholesterol in liver and plasma were significantly lower than age matched FF mice (Fig. 1Q,R). Associated with the lower levels of liver and circulating triglyceride and cholesterol, mRNA levels of lipogenesis-related genes such as fatty acid synthase (*Fasn*), diacylglycerol O-acyltransferase 1 (*Dgat1*), and sterol regulatory element-binding protein 1 (*Srebf1*) were significantly decreased in *Nestin-Lgr4⁻/⁻* mice. Hepatic genes related to β-oxidation like adipose triglyceride lipase (*Atgl*), or carnitine palmitoyl transferase 1 (*Cpt1α*) were higher, whereas genes related to lipid transport remained unchanged (Fig. 1S). These results indicate that deficiency of LGR4 in nestin neurons can ameliorate liver steatosis.

The adipose tissue is a central metabolic organ in the regulation of whole-body energy homeostasis. BAT and SAT are recognized as major sites of thermogenesis in controlling whole-body energy expenditure and body fat mass. Our results showed that tissue weights of SAT and BAT in *Nestin-Lgr4⁻/⁻* mice fed HFD were significantly lower than FF mice (Figs. 1D and EV1C). Adipocyte size and lipid droplets in brown and white adipocytes were smaller (Fig. 1P,V). Further, deficiency of LGR4 in nestin neurons significantly increases

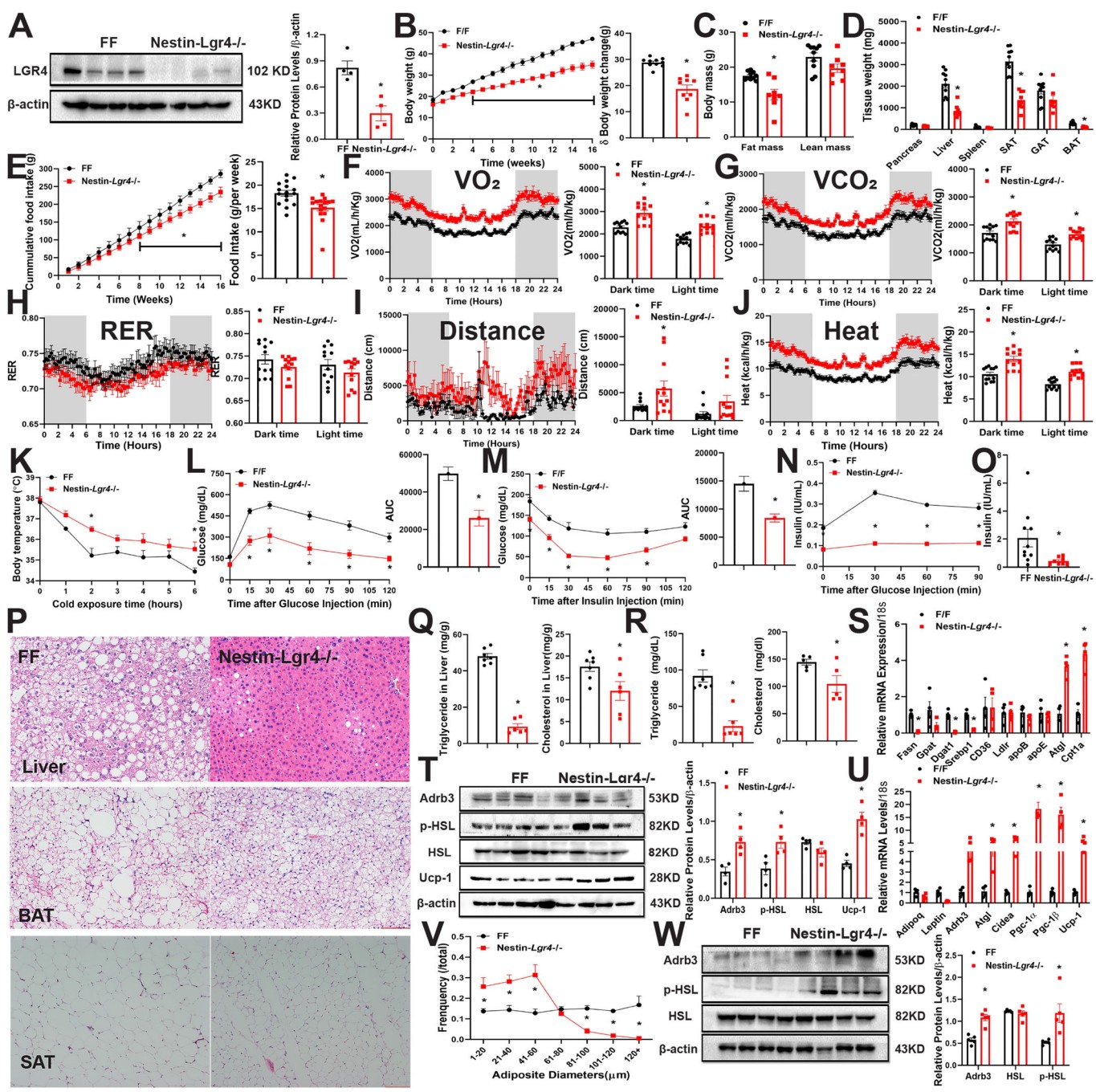

Ucp-1 protein and mRNA levels in BAT relevant to FF mice. Also increased were other thermogenic genes such as *Atgl*, *Cidea*, *Pgc1α* and *Pgc1β* in BAT (Fig. 1T,U). In contrast, mRNA levels of adiponectin (*Adipoq*) and *leptin* were significantly reduced (Fig. 1U). Phosphorylation of hormone sensitive lipase (p-HSL) in both BAT and SAT of *Nestin-Lgr4$^{-/-}$* mice were significantly increased relative to FF mice, consistent with an increment of lipolysis in both BAT and SAT. In addition, protein levels of Adrb3 in both BAT and SAT were higher in *Nestin-Lgr4$^{-/-}$* mice (Fig. 1T,W). These results suggest that deficiency of LGR4 in nestin neurons may promote thermogenesis by increasing lipolysis in BAT and SAT in a manner dependent on sympathetic activation.

## Knockdown of LGR4 in *Sp1* specific neurons reduces HFD-induced adiposity and improves whole body metabolism by increasing energy expenditure and suppressing food intake

Since nestin is predominantly expressed in the early stage of neuronal development, we next examined the metabolic benefits of LGR4 deficiency in mature neurons using the synaptophysin 1 (*Sp1*)-*Lgr4$^{-/-}$* transgene. Specific deletion of LGR4 in mature neurons was achieved by cross-breeding the *Sp1-Cre* mice with *Lgr4$^{flox/flox}$* animals. Deficiency of LGR4 in the whole brain was confirmed by western blotting (Fig. 2A). The body weight and body composition in *Sp1-cre* mice and FF mice

◄ **Figure 1. Knockdown of *Lgr4* in *nestin* neurons renders mice resistant to high fat diet-induced obesity by increasing energy expenditure and concurrently inhibiting food intake.**

Male mice at 6 weeks old were fed HFD for 16 weeks. All data were shown as mean±SEM, differences between two groups were analyzed by two-tailed Student's *t* test., *$P < 0.05$. (A) Whole brain extracts were immunoblotted with anti-LGR4 or β-actin, FF $n = 4$, *Nestin-Lgr4*$^{-/-}$ $n = 4$ ($P = 0.0043$). (B) Body weight curves (left, $P < 0.05$ for week 8 to 16) and body weight changes (right, $P < 0.0001$, FF $n = 10$, *Nestin-Lgr4*$^{-/-}$ $n = 9$. (C) Body composition: fat mass and lean mass contents measured by MRI, FF $n = 10$, *Nestin-Lgr4*$^{-/-}$ $n = 8$ (fat mass: $P = 0.0001$). (D) Tissue weights, FF $n = 10$, *Nestin-Lgr4*$^{-/-}$ $n = 8$ ($P < 0.0001$ for liver, SAT and BAT). (E) Cumulative food intake (left, $P < 0.05$ for week 8 to 16) and weekly food intake (right, $P = 0.002$), FF $n = 10$, *Nestin-Lgr4*$^{-/-}$ $n = 8$. (F–J) TSE phenotype data, mice were placed in TSE chamber for 7 days, data from last 3 days were recorded and analyzed, FF $n = 4$, Nestin-Lgr4$-/-$ $n = 4$. (F) $O_2$ consumption (dark time: $P < 0.0001$; light time: $P < 0.0006$). (G) $CO_2$ production (dark time: $P = 0.0001$; light time: $P = 0.0006$). (H) Respiratory exchange ratio (RER). (I) Locomotor activity (dark time: $P = 0.0363$). (J) Energy expenditure (dark time: $P < 0.0001$; light time: $P = 0.0001$). (K) Rectal body temperature under 4 °C, FF $n = 6$, *Nestin-Lgr4*$^{-/-}$ $n = 7$ (2 h: $P = 0.0077$). (L) IPGTT, FF $n = 10$, *Nestin-Lgr4*$^{-/-}$ $n = 8$ (AUC: $P = 0.0005$). (M) IPITT, FF $n = 10$, *Nestin-Lgr4*$^{-/-}$ $n = 8$ (AUC: $P = 0.0016$). (N) Insulin secretion after glucose injection, FF $n = 6$, *Nestin-Lgr4*$^{-/-}$ $n = 6$ ($P < 0.0001$ for all time points). (O) Plasma Insulin, FF $n = 10$, *Nestin-Lgr4*$^{-/-}$ $n = 8$ ($P = 0.0339$). (P) Representative H&E staining of liver (scale bar: 500 μm), BAT (scale bar: 500 μm) and SAT (scale bar: 500 μm) sections, scaler bar, 100 pixels. (Q) Triglyceride content (left, $P < 0.0001$) and total cholesterol content (right, $P = 0.034$) in liver, FF $n = 7$, *Nestin-Lgr4*$^{-/-}$ $n = 6$. (R) Triglyceride content (left, FF $n = 7$, *Nestin-Lgr4*$^{-/-}$ $n = 6$, $P < 0.0001$) and total cholesterol content (right, FF $n = 5$, *Nestin-Lgr4*$^{-/-}$ $n = 5$, $P = 0.0393$) in plasma. (S) mRNA levels of genes related to lipid genesis, transport and β-oxidation in liver tissue, FF $n = 4$, *Nestin-Lgr4*$^{-/-}$ $n = 4$ (Fasn: $P = 0.00067$; Dgat1: $P = 0.001312$; Srebp1: $P = 0.002056$; Atgl: $P = 0.002056$; Cpt1a: $P = 0.002056$). (T) BAT extracts were immunoblotted with indicated antibodies: anti-Adrb3 ($P = 0.0019$), HSL, p-HSL ($P = 0.0052$), UCP-1 ($P < 0.0001$) and β-actin, FF $n = 4$, *Nestin-Lgr4*$^{-/-}$ $n = 4$. (U) Expression of genes related to thermogenesis and browning in BAT, FF $n = 4$, *Nestin-Lgr4*$^{-/-}$ $n = 4$ (Atgl: $P = 0.0419$; Cidea: $P = 0.0043$; Pgc-1α: $P < 0.0001$; Pgc-1β: $P < 0.0001$; Ucp-1: $P = 0.0144$). (V) Adipocyte size of SAT, FF $n = 6$, *Nestin-Lgr4*$^{-/-}$ $n = 5$ (1–20 μm: $P = 0.011$; 20–40 μm: $P = 0.0023$; 40–60 μm: $P < 0.0001$; 80–100 μm: $P < 0.0001$; 100–120 μm: $P = 0.0114$; 120 + μm: $P = 0.0002$). (W) SAT extracts were immunoblotted with indicated antibodies: anti-Adrb3 ($P = 0.0318$), HSL, p-HSL ($P = 0.0046$), and β-actin, FF $n = 4$, *Nestin-Lgr4*$^{-/-}$ $n = 4$. Source data are available online for this figure.

were comparable (Appendix Fig. S2A). Both male and female *Sp1-Lgr4*$^{-/-}$ mice fed NCD demonstrated no significant difference in body weight, body composition, tissue weights and food intake (Appendix Fig. S2B–E), glucose tolerance and insulin sensitivity (Appendix Fig. S2F,G) relative to sex and age matched FF mice. When fed 60% HFD for 16 weeks, *Sp1-Lgr4*$^{-/-}$ mice demonstrated significantly lower body weight, fat mass, tissue weights of liver, SAT and BAT relative to sex and age matched FF mice (Fig. 2B–D; Appendix Fig. S2H–J). Cumulative and weekly food intake in male *Sp1-Lgr4*$^{-/-}$ mice were significantly reduced relative to FF mice, although female *Sp1-Lgr4*$^{-/-}$ mice only showed moderate reduction (Fig. 2E; Appendix Fig. S2K). $O_2$ consumption and $CO_2$ production in dark and light phase were significantly increased in *Sp1-Lgr4*$^{-/-}$ mice relative to FF mice, whereas RER was reduced (Fig. 2F–H; Appendix Fig. S6D,E). Activity and heat production were also increased in the transgenes (Fig. 2I,J; Appendix Fig. S6F). Core body temperature in both male and female *Sp1-Lgr4*$^{-/-}$ mice were higher than in FF mice under 4 °C cold exposure (Fig. 2K; Appendix Fig. S2L). These results indicate an increment in energy expenditure in *Sp1-Lgr4*$^{-/-}$ mice. Along with lean phenotype, glucose tolerance and insulin sensitivity in *Sp1-Lgr4*$^{-/-}$ mice were significantly improved relative to FF mice (Fig. 2L,M; Appendix Fig. S2M,N). Levels of plasma insulin were lower (Fig. 2N; Appendix Fig. S2O). HFD-induced liver steatosis was significantly less in the *Sp1-Lgr4*$^{-/-}$ mice. Hepatocyte lipid droplets were smaller and less abundant relative to FF mice (Fig. 2O). Hepatic and plasma levels of triglyceride and cholesterol were significantly lower than age matched FF mice (Fig. 2P,Q). mRNA levels of lipogenesis-relevant genes such as *Fasn*, *Dgat1* and *Srebp1* were significantly lower than FF mice, whereas β-oxidation related genes like *Atgl* and *Cpt1α* were significantly higher (Fig. 2R).

Moreover, lipid droplets and adipocyte size in both BAT and SAT of *Sp1-Lgr4*$^{-/-}$ mice were smaller than FF mice (Fig. 2O,U). For BAT, both Ucp-1 and Adrb3 protein and mRNA levels were significantly increased, while pHSL protein levels in BAT and Adrb3 mRNA levels in SAT were higher than FF mice (Fig. 2O,S,T). Other thermogenesis related genes such as *Atgl*, *Cidea*, *Pgc1α* and *Pgc1β* were significantly increased in both BAT and SAT, while white adipose genes including *Adipoq* and *leptin* were decreased (Fig. 2T,V). These results indicate that deficiency of LGR4 in Sp1 neurons may promote energy expenditure by increasing lipolysis in adipose tissues.

## Knockdown of LGR4 in *AgRP* specific neurons decreases food intake and increases energy expenditure, leading to reduction of in mice fed HFD

Under physiological conditions, energy balance is controlled by the hypothalamus. To further explore the specific neurons responsible for the metabolic benefits of hypothalamic LGR4 deficiency, we generated transgenes with conditional deletion of LGR4 in ARC neurons. *Pomc-Lgr4*$^{-/-}$ mice were generated by crossbreeding *Lgr4*$^{flox/flox}$ mice with *Pomc-Cre* transgenes. Deficiency of LGR4 in POMC neurons demonstrated no significant alterations in food intake, body weight, body composition or other related metabolic consequences relative to FF mice fed either NCD or HFD (Appendix Fig. S3B–M). On the other hand, *Agrp-Lgr4*$^{-/-}$ mice generated by crossbreeding *Lgr4*$^{flox/flox}$ mice with *Agrp-ERT-Cre* animals (Fig. 3A) demonstrated significant alterations in food intake, body weight, body composition, liver weight and fat mass relative to FF control mice in the condition of HFD feeding (Fig. 3B–E), but not with NCD feeding (Appendix Fig. S4B–D). These alterations were present in male mice but not in female transgenes (Fig. EV2F–I). In addition, energy expenditure and body temperature under 4 °C cold exposure were higher in *Agrp-Lgr4*$^{-/-}$ mice relative to sex and age matched FF control animals (Figs. 3F–K and EV2J–N,Q; Appendix Fig. S6G–I). Consistent with the lean phenotype, male *Agrp-Lgr4*$^{-/-}$ mice showed an improvement in glucose tolerance and insulin sensitivity, as well as a reduction in plasma insulin levels relative to FF mice (Figs. 3L–N and EV2O,P). Deficiency of LGR4 in AgRP neurons significantly ameliorated HFD-induced liver steatosis, as evidenced by reduction in the size and number of lipid droplets, and triglyceride and cholesterol levels in both liver and plasma relative to FF control littermates (Fig. EV2A–C). In BAT, lipid droplets were smaller, levels of thermogenesis-related proteins such as Adrb3, p-HSL and Ucp-1 were higher than in FF control littermates (Fig. 3O,P). Similarly, lipid droplets and adipocyte size of SAT were decreased relative to FF control mice (Fig. EV2D,E). Thus, our data indicate knockdown of LGR4 in Agrp specific neurons resistant to HFD induced obesity by increasing energy expenditure and reducing food intake.

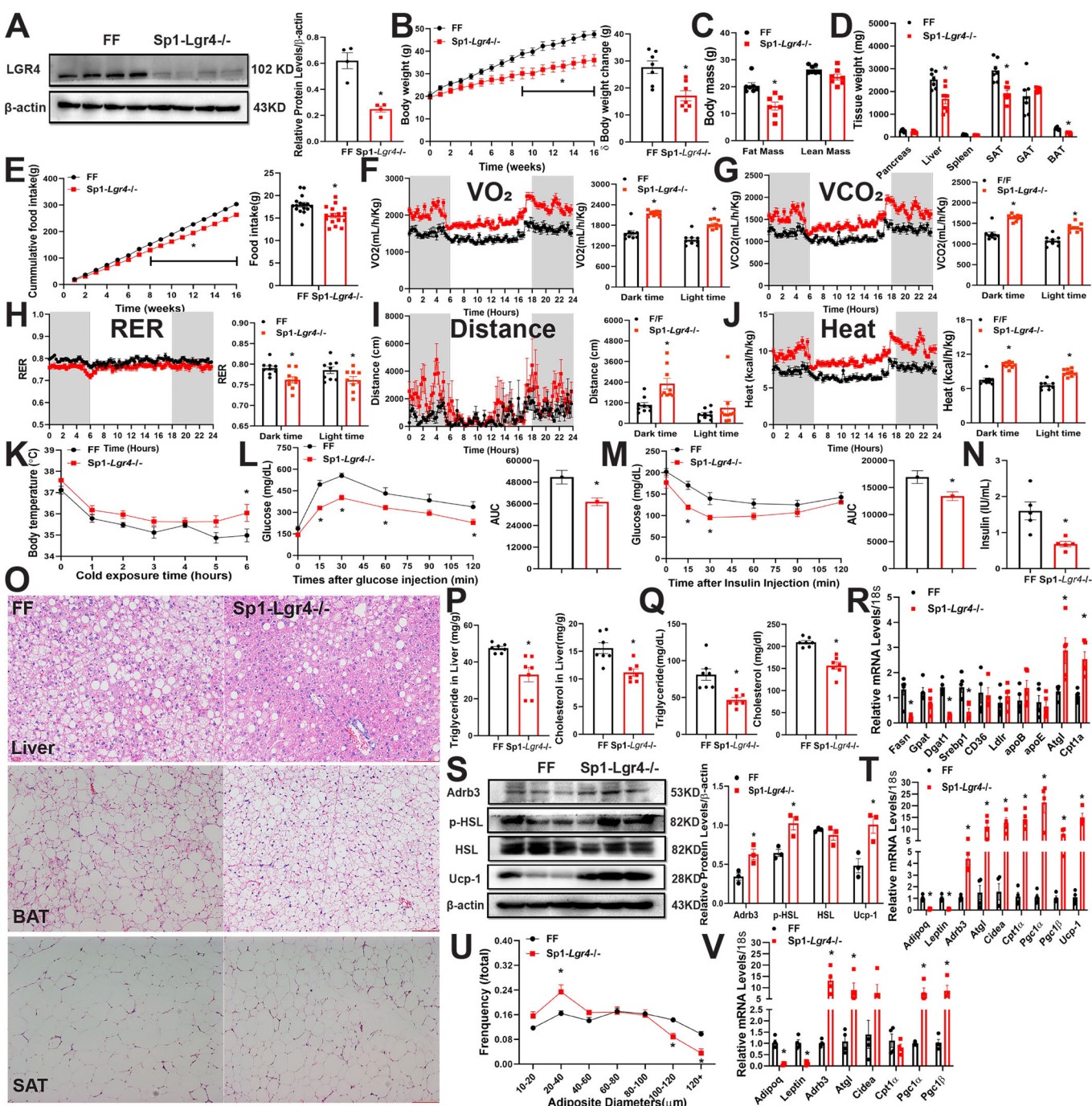

## Knockdown of LGR4 in *Sf1* neurons reduces adiposity, increases energy expenditure and improves organism metabolism in mice fed HFD

To explore the role of LGR4 signaling in VMH nuclei of the hypothalamus, we generated the *Sf1-Lgr4⁻/⁻* mice by crossbreeding *Lgr4flox/flox* (FF) mice with *Sf1-cre* transgene. Western blotting results revealed a deficiency of LGR4 in hypothalamus derived from *Sf1-Lgr4⁻/⁻* mice (Fig. 4A). Body weight, body composition in *Sf1-Cre* mice were comparable with FF mice fed NCD or HFD (Appendix Fig. S5A).

Deficiency of LGR4 in Sf1 neurons demonstrated no effect on body weight, body mass composition, food intake, tissue weights, glucose tolerance and insulin tolerance in mice fed NCD (Appendix Fig. S5B–G). However, body weight and body weight changes in *Sf1-Lgr4⁻/⁻* mice fed HFD for 16 weeks were dramatically lower than age and sex matched FF control littermates (Figs. 4B and EV3H). Also reduced were the fat mass (Figs. 4C and EV3I), as well as the tissue weights of liver, BAT and SAT (Figs. 4D and EV3J). O₂ consumption and CO₂ production, activity and energy expenditure in *Sf1-Lgr4⁻/⁻* mice were higher than age and sex matched FF mice (Figs. 4F–J

◄ **Figure 2. Knockdown of LGR4 in *Sp1* specific neurons reduces adiposity and food intake, increases energy expenditure and affects whole body metabolism in mice fed HFD.**

Male mice were fed on HFD at 6 weeks old for 16 weeks. All data were shown as mean ± SEM, differences between two groups were analyzed by two-tailed Student's *t* test. Comparisons between more than two groups or variables were analyzed by one-way or two-way ANOVA and/or Tukey's post hoc test, *$P < 0.05$. (A) Whole brain extracts were immunoblotted with aniti-LGR4 or β-actin, FF $n = 4$, *Sp1-Lgr4$^{-/-}$* $n = 4$, $P = 0.0013$. (B) Body weight curves (left, $P < 0.05$ for week 9 to 16) and body weight changes (right, $P = 0.0038$), FF $n = 7$, *Sp1-Lgr4$^{-/-}$* $n = 7$. (C) Body composition: FF $n = 7$, *Sp1-Lgr4$^{-/-}$* $n = 7$ (fat mass: $P = 0.0002$). (D) Tissue weights, FF $n = 7$, *Sp1-Lgr4$^{-/-}$* $n = 7$ (liver: $P = 0.014911$; SAT: $P = 0.001177$; BAT: $P = 0.001177$). (E) Cumulative food intake (left, $P < 0.05$ for week 10 to 16) and weekly food intake (right, $P = 0.0024$), FF $n = 7$, *Sp1-Lgr4$^{-/-}$* $n = 7$. (F–J) TSE phenotype data, mice were placed in TSE chamber for 7 days, data from last 3 days were recorded and analyzed, FF $n = 3$, *Sp1-Lgr4$^{-/-}$* $n = 3$. (F) $O_2$ consumption (dark time: $P < 0.0001$; light time: $P < 0.0001$). (G) $CO_2$ production (dark time: $P < 0.0001$, light time: $P < 0.0001$). (H) Respiratory exchange ratio (RER, dark time: $P = 0.0256$). (I) Locomotor activity (dark time: $P = 0.0044$). J Energy expenditure (dark time: $P < 0.0001$, light time: $P < 0.0001$). (K) Rectal body temperature under 4 °C, FF $n = 6$, *Sp1-Lgr4$^{-/-}$* $n = 5$ (6 h: $P = 0.0100$). (L, M) IPGTT (L, AUC: $P = 0.0085$) and IPITT (M, AUC: $P = 0.0291$), FF $n = 7$, *Sp1-Lgr4$^{-/-}$* $n = 7$ (N) Plasma insulin, FF $n = 5$, *Sp1-Lgr4$^{-/-}$* $n = 5$ ($P = 0.0074$). (O) Representative H&E staining of liver (scale bar: 500 μm), BAT (scale bar: 500 pixel) and SAT (scale bar: 500 pixel) sections. (P) Triglyceride content (left, $P = 0.0035$) and total cholesterol content (right, $P = 0.0030$) in liver, FF $n = 7$, *Sp1-Lgr4$^{-/-}$* $n = 7$. (Q) Triglyceride content (left, $P = 0.0017$) and total cholesterol content (right, $P < 0.0001$) in plasma, FF $n = 7$, *Sp1-Lgr4$^{-/-}$* $n = 7$. (R) Expression of genes related to lipid genesis, transport and β-oxidation in liver tissue, FF $n = 4$, *Sp1-Lgr4$^{-/-}$* $n = 4$ (Fasn: $P = 0.0463$, Gpat1: $P = 0.0238$, Atgl: $P < 0.0001$, Cpt1α: $P < 0.0001$). (S) BAT extracts were immunoblotted with indicated antibodies: anti-Adrb3 ($P = 0.0466$), HSL, p-HSL ($P = 0.0067$), UCP-1 ($P = 0.0003$) and β-actin, FF $n = 3$, *Sp1-Lgr4$^{-/-}$* $n = 3$. (T) Expression of genes related to thermogenesis and browning in BAT, FF $n = 4$, *Sp1-Lgr4$^{-/-}$* $n = 4$ (Adipoq: $P = 0.003390$; Leptin: $P = 0.003390$; Adrb3: $P = 0.005268$; Atgl: $P = 0.009156$; Cidea: $P = 0.009156$; Cpt1α: $P = 0.003390$; Pgc-1α: $P = 0.009156$; Pgc-1β: $P = 0.006246$; Ucp-1: $P = 0.002289$). (U) Adipocyte size of SAT, FF $n = 7$, *Sp1-Lgr4$^{-/-}$* $n = 5$ (20–40 μm: $P = 0.0004$; 100–120 μm: $P = 0.0080$; 120 + μm: $P = 0.0020$). (V) Expression of genes related to thermogenesis and browning in SAT, FF $n = 4$, *Sp1-Lgr4$^{-/-}$* $n = 4$ (Adipoq: $P = 0.000392$; Leptin: $P = 0.000779$; Adrb3: $P = 0.004584$; Atgl: $P = 0.046944$; Pgc-1α: $P = 0.021509$; Pgc-1β: $P = 0.020590$). Source data are available online for this figure.

and EV3L–P; Appendix Fig. 6J–L). Food intake was comparable relative to age and sex matched FF mice (Figs. 4E and EV3K). The body temperature in male *Sf1-Lgr4$^{-/-}$* mice under 4 °C cold exposure were slightly higher than FF mice, although the female mice showed comparable body temperature (Figs. 4K and EV3Q).

Analysis of glucose metabolism showed a significant improvement in glucose and insulin tolerance in *Sf1-Lgr4$^{-/-}$* mice relative to age and sex matched FF control littermates (Fig. 4L,M and EV3R,S). Plasma levels of insulin were significantly reduced (Figs. 4N and EV3T). Liver steatosis induced by HFD was decreased relative to FF control mice. Lipid droplet size and numbers, levels of triglyceride and cholesterol, mRNA levels of lipogenesis-related genes like *Fasn*, *Dgat1* and *Srebp1* were significantly reduced in the liver. Hepatic levels of β-oxidation-related genes such as *Cpt1α* and *Atgl* were increased (Fig. EV3A,B,D). In addition, plasma levels of triglyceride in *Sf1-Lgr4$^{-/-}$* mice were significantly lower (Fig. EV3C). In BAT, lipid droplets were smaller, whereas mRNA and protein expression levels of Adrb3 and Ucp-1, protein levels of p-HSL, mRNA levels of thermogenesis related genes such as *Cidea*, *Pgc1α* and *Pgc1β* were significantly increased (Fig. 4O–Q). In SAT, numbers and size of lipid droplets, as well as adipocyte size were significantly reduced. Thermogenesis-related genes like *Cidea*, *Pgc1α* and *Pgc1β* were increased, while *Adipoq* and *leptin* were decreased (Fig. EV3E–G). These results indicate that deficiency of LGR4 in Sf1 neurons renders mice resistant to diet induced obesity by increasing energy expenditure but not effecting food intake.

## Intrascapular BAT bilateral sympathectomy reverses the metabolic benefits in *Agrp-Lgr4$^{-/-}$* mice fed HFD

The sympathetic system affects metabolic homeostasis by orchestrating the activity of metabolic organs such as the pancreas, liver, white and brown adipose tissues (Martinez-Sanchez et al, 2022). Central sympathetic nerves innervate BAT to control thermogenesis and energy balance (Saito, 2013). To determine whether LGR4 in AgRP neurons controls organism metabolism via sympathetic output to BAT, we performed intrascapular BAT bilateral sympathectomy on mice fed HFD for 8 weeks. Intrascapular BAT bilateral sympathectomy eliminated

protection from diet-induced obesity in *Agrp-Lgr4$^{-/-}$* mice. Body weight and fat mass were higher than *Agrp-Lgr4$^{-/-}$* mice with sham surgery. Body weight and body composition in FF mice with intrascapular BAT bilateral sympathectomy and sham surgery were comparable (Fig. 5A,B). Food intake in *Agrp-Lgr4$^{-/-}$* mice with intrascapular BAT bilateral sympathectomy was higher than *Agrp-Lgr4$^{-/-}$* mice with sham surgery (Fig. 5D). Consistent with the change of body weight, intrascapular BAT bilateral sympathectomy significantly increased the tissue weights of BAT, SAT, GAT and liver relative to *Agrp-Lgr4$^{-/-}$* mice with sham surgery. Tissue weights of FF mice with sympathectomy or sham surgery were comparable (Fig. 5C). Intrascapular BAT bilateral sympathectomy reversed the increment in the activity, energy expenditure, $O_2$ consumption and $CO_2$ production normalized to body weight in *Agrp-Lgr4$^{-/-}$* mice (Fig. 5E–I; Appendix Fig. S6M–O). These results indicate that intrascapular BAT bilateral sympathectomy reverses the decrement of food intake and increment of energy expenditure induced by LGR4 deficiency in AgRP neurons. Further, intrascapular BAT bilateral sympathectomy significantly attenuated the improvement in glucose and lipid metabolism in *Agrp-Lgr4$^{-/-}$* mice (Fig. 5J,K). The reduction of plasma insulin levels in *Agrp-Lgr4$^{-/-}$* mice was also reversed (Fig. 5L). In addition, intrascapular BAT bilateral sympathectomy significantly attenuated the improvement in HFD-induced liver steatosis in *Agrp-Lgr4$^{-/-}$* mice as demonstrated by (1) a marked increase of lipid droplet size and numbers in liver; (2) higher levels of triglyceride and cholesterol in both liver and plasma; 3) higher expression levels of lipogenesis related gene like *Dgat1* and lower expression levels of lipid oxidation genes such as *Atgl* and *Cpt1α* in liver. Intrascapular BAT bilateral sympathectomy demonstrated no significant effect on liver steatosis in FF mice (Fig. EV4A–D). Intrascapular BAT bilateral sympathectomy also reversed the effects of *Agrp-Lgr4$^{-/-}$* on adipose lipolysis in both BAT and SAT. The alterations in lipid droplets, individual adipocyte size, mRNA and protein levels of Ucp-1 and Adrb3, thermogenesis-related genes like *Adrb3*, *Cidea*, *Pgc1α* and *Pgc1β* were reversed. In contrast, intrascapular BAT bilateral sympathectomy demonstrated no effect on thermogenesis-related molecules in FF mice (Figs. 5M–O and EV4I–K).). In summary, intrascapular BAT bilateral sympathectomy reverses the metabolic benefits of *Agrp-Lgr4$^{-/-}$* mice.

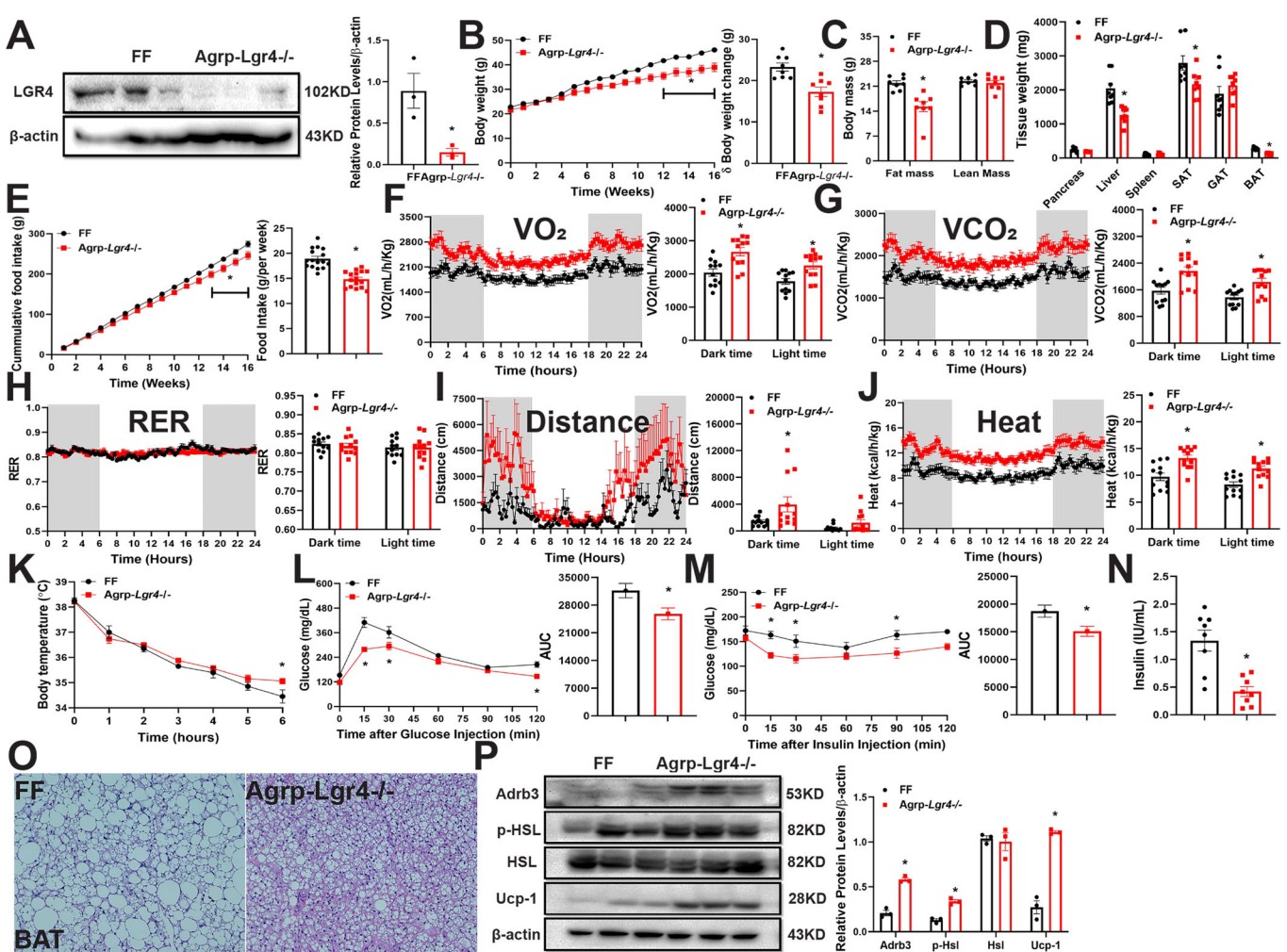

**Figure 3. Knockdown of LGR4 in *AgRP* specific neurons decreases food intake and increases energy expenditure, leading to reduction of adiposity in male mice fed HFD.**

Male mice were fed on HFD, beginning at 6 weeks old for 16 weeks. All data were shown as mean ± SEM, differences between two groups were analyzed by two-tailed Student's *t* test. Comparisons between more than two groups or variables were analyzed by one-way or two-way ANOVA and/or Tukey's post hoc test, *$P < 0.05$. (A) Hypothalamic extracts were immunoblotted with aniti-LGR4 or β-actin, FF $n = 3$, *Agrp-Lgr4*$^{-/-}$ $n = 3$ ($P = 0.0050$). (B) Body weight curves (left, $P < 0.05$ for week 12 to 16) and body weight changes (right, $P = 0.0013$), FF $n = 8$, *Agrp-Lgr4*$^{-/-}$ $n = 8$. (C) Body composition FF $n = 8$, *Agrp-Lgr4*$^{-/-}$ $n = 8$ (fat mass: $P = 0.000656$). (D) Tissue weights, FF $n = 8$, *Agrp-Lgr4*$^{-/-}$ $n = 8$ (liver: $P = 0.000655$; SAT: $P = 0.028331$; BAT: $P < 0.0001$). (E) Cumulative food intake (left, $P < 0.05$ for week 12 to 16) and weekly food intake (right, $P < 0.0001$), FF $n = 8$, *Agrp-Lgr4*$^{-/-}$ $n = 8$. (F–J) TSE phenotype data, mice were placed in TSE chamber for 7 days, data for the last 3 days were recorded and analyzed, FF $n = 4$, *Agrp-Lgr4*$^{-/-}$ $n = 4$. (F) O$_2$ consumption (dark time: $P = 0.0267$; light time: $P = 0.0541$). (G) CO$_2$ production (dark time: $P = 0.0058$; light time: $P = 0.0141$). (H) Respiratory exchange ratio (RER). (I) Locomotor activity (dark time: $P = 0.0115$). (J) Energy expenditure (dark time: $P = 0.0002$; light time: $P = 0.0006$). (K) Rectal body temperature under 4 °C, FF $n = 5$, *Agrp-Lgr4*$^{-/-}$ $n = 5$ (6 h: $P = 0.0076$). (L, M) IPGTT (L, AUC: $P = 0.0251$) and IPITT (M, AUC: $P = 0.0224$), FF $n = 8$, *Agrp-Lgr4*$^{-/-}$ $n = 8$. (N) Plasma insulin, FF $n = 8$, *Agrp-Lgr4*$^{-/-}$ $n = 8$ ($P = 0.0006$). (O) Representative H&E staining of BAT (scale bar: 500 pixel) sections. (P) BAT extracts were immunoblotted with indicated antibodies: anti-Adrb3 ($P = 0.0002$), HSL, p-HSL ($P = 0.0274$), UCP-1 ($P < 0.0001$) and β-actin, FF $n = 3$, *Agrp-Lgr4*$^{-/-}$ $n = 3$. Source data are available online for this figure.

## Intrascapular BAT bilateral sympathectomy reverses the metabolic benefits of LGR4 deficiency in *Sf1* neurons

To determine whether sympathetic output to BAT contributes to the metabolic benefits of LGR4 deficiency in Sf1 neurons, we performed intrascapular BAT bilateral sympathectomy on male *Sf1-Lgr4*$^{-/-}$ mice fed HFD for 8 weeks. Intrascapular BAT bilateral sympathectomy increased diet-induced obesity in *Sf1-Lgr4*$^{-/-}$ mice but not in FF control littermates relative to sham mice. Body weight, fat mass, tissue weights of liver, BAT and SAT in *Sf1-Lgr4*$^{-/-}$

mice with intrascapular BAT bilateral sympathectomy were higher than in *Sf1-Lgr4*$^{-/-}$ mice with sham surgery (Fig. 6A–C). Body weight, fat mass and tissue weights of FF control mice with intrascapular BAT bilateral sympathectomy and sham surgery were comparable (Fig. 6A–C). Glucose tolerance and insulin sensitivity in *Sf1-Lgr4*$^{-/-}$ mice with intrascapular BAT bilateral sympathectomy were significantly different relative to *Sf1-Lgr4*$^{-/-}$ mice with sham surgery (Fig. 6E,F). Along with this, plasma insulin levels in *Sf1-Lgr4*$^{-/-}$ mice with intrascapular BAT bilateral sympathectomy were higher relative to the *Sf1-Lgr4*$^{-/-}$ transgenes with sham

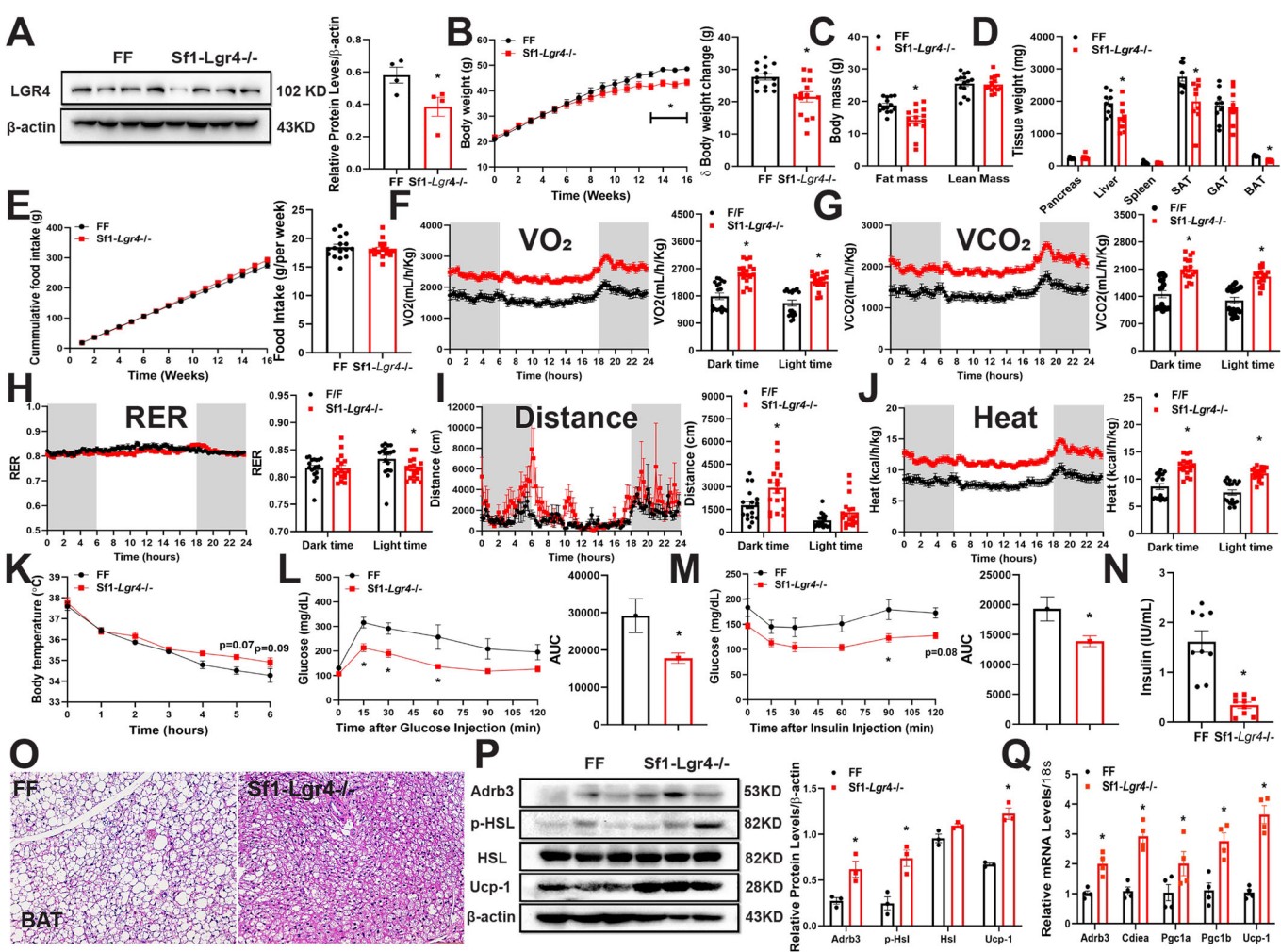

**Figure 4. Knockdown of LGR4 in *Sf1* neurons reduces adiposity, increases energy expenditure in mice fed HFD.**

Male mice were fed on HFD, beginning at 6 weeks old for 16 weeks. All data were shown as mean ± SEM, differences between two groups were analyzed by two-tailed Student's *t* test. Comparisons between more than two groups or variables were analyzed by one-way or two-way ANOVA and/or Tukey's post hoc test, *$P < 0.05$. **(A)** Hypothalamic extracts were immunoblotted with aniti-LGR4 or β-actin, FF $n = 4$, *Sf1-Lgr4$^{-/-}$* $n = 4$ ($P = 0.0438$). **(B)** Body weight curves (left, $P < 0.05$ for week 13–16) and body weight changes (right, $P = 0.0026$), FF $n = 14$, *Sf1-Lgr4$^{-/-}$* $n = 14$. **(C)** Body composition FF $n = 14$, *Sf1-Lgr4$^{-/-}$* $n = 14$ (fat mass: $P = 0.0006$). **(D)** Tissue weights, FF $n = 9$, *Sf1-Lgr4$^{-/-}$* $n = 9$ (liver: $P = 0.028774$; SAT: $P = 0.008378$; BAT: $P < 0.0001$). **(E)** Cumulative food intake (left) and weekly food intake (right), FF $n = 14$, *Sf1-Lgr4$^{-/-}$* $n = 14$. **(F–J)** TSE phenotype data, FF $n = 6$, *Sf1-Lgr4$^{-/-}$* $n = 6$, mice were placed in TSE chamber for 7 days, data were recorded in the last 3 days and analyzed. **(F)** O$_2$ consumption (dark time: $P < 0.0001$; light time: $P < 0.0001$). **(G)** CO$_2$ production (dark time: $P < 0.0001$; light time: $P < 0.0001$). **(H)** Respiratory exchange ratio (RER). **(I)** Locomotor activity (dark time: $P = 0.010245$; light time: $P = 0.034663$). **(J)** Energy expenditure (dark time: $P < 0.0001$; light time: $P < 0.0001$). **(K)** Rectal body temperature under 4 °C, FF $n = 9$, *Sf1-Lgr4$^{-/-}$* $n = 8$ (5 h: $P = 0.07$; 6 h: $P = 0.08$). **(L, M)** IPGTT (**L**, $P = 0.0293$) and IPITT (**M**, $P = 0.0263$), FF $n = 9$, *Sf1-Lgr4$^{-/-}$* $n = 9$. **(N)** Plasma Insulin ($P < 0.0001$), FF $n = 9$, *Sf1-Lgr4$^{-/-}$* $n = 9$. **(O)** Representative H&E staining of BAT (scale bar: 500 pixel) sections. **(P)** BAT extracts were immunoblotted with indicated antibodies: anti-Adrb3 ($P = 0.0017$), HSL, p-HSL ($P < 0.0001$), UCP-1 ($P < 0.0001$) and β-actin, $n = 3$. **(Q)** Expression of genes related to thermogenesis and browning in BAT, FF $n = 4$, *Sf1-Lgr4$^{-/-}$* $n = 4$ (Adrb3: $P = 0.0147$; Cidea: $P < 0.0001$; Pgc-1α: $P = 0.0147$; Pgc-1β: $P = 0.0001$; Ucp-1: $P < 0.0001$). Source data are available online for this figure.

surgery, while insulin levels in FF control littermates remained unaltered (Fig. 6G). Before and after surgery, food intake in *Sf1-Lgr4$^{-/-}$* mice was similar (Fig. 6D). This observation indicates that intrascapular BAT bilateral sympathectomy reverses the lean phenotype of *Sf1-Lgr4$^{-/-}$* mice without changing food intake. Further, hepatocyte lipid droplets in *Sf1-Lgr4$^{-/-}$* mice with intrascapular BAT bilateral sympathectomy were bigger and more abundant than *Sf1-Lgr4$^{-/-}$* mice with sham surgery. Also reversed were reduction of hepatic and plasma levels of triglyceride and cholesterol, mRNA expression levels of lipid genesis related gene

*Dgat-1*. The increment of the β-oxidation gene *Atgl* in *Sf1-Lgr4$^{-/-}$* mice was significantly attenuated by intrascapular BAT bilateral sympathectomy (Fig. EV4E–H). These results indicate that intrascapular BAT bilateral sympathectomy attenuates the amelioration of HFD-induced liver steatosis in *Sf1-Lgr4$^{-/-}$* mice. Further, reduction in the lipid content of both BAT and SAT, were attenuated by intrascapular BAT bilateral sympathectomy. Increases in mRNA and protein levels of Ucp-1, Adrb3 and p-HSL in BAT were reversed (Figs. 6H–J and EV4M,N). In addition, the expression of thermogenesis related genes like *Cidea*, *Pgc-1α* and

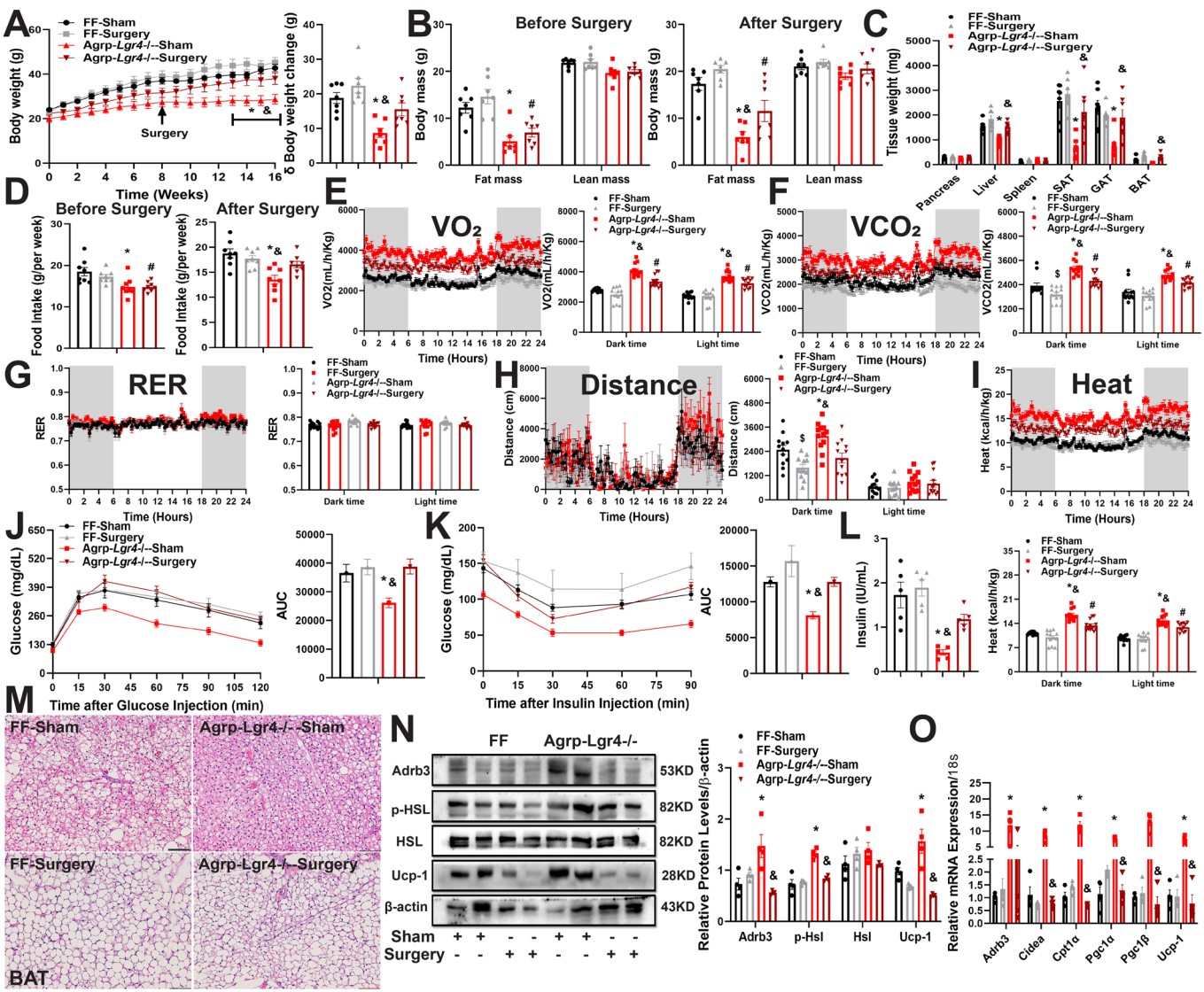

**Figure 5. Intrascapular BAT bilateral sympathectomy reverses the lean phenotype and the metabolic benefits in *Agrp-Lgr4⁻/⁻* mice fed HFD.**

Intrascapular BAT Bilateral Sympathectomy was performed on male mice fed on HFD for 8 weeks, beginning at 6 weeks. All data were shown as mean ± SEM, comparisons between more than two groups or variables were analyzed by one-way or two-way ANOVA and/or Tukey's post hoc test, $P < 0.05$, *FF-sham vs *Agrp-Lgr4⁻/⁻*-sham, $^\$$FF-sham vs FF-Surgery, $^\&$*Agrp-Lgr4⁻/⁻*-sham vs *Agrp-Lgr4⁻/⁻*-Surgery, #FF- Surgery vs *Agrp-Lgr4⁻/⁻*-Surgery. (A) Body weight curve (left, *$P < 0.05$ for week 4 to 16; $^\&P < 0.05$ for week 12–16), body weight change after surgery (right. *$P = 0.0023$; $^\&P = 0.0499$), FF-sham $n = 7$, *Agrp-Lgr4⁻/⁻*-sham $n = 7$, FF-Surgery $n = 7$, *Agrp-Lgr4⁻/⁻*-Surgery $n = 7$. (B) Body composition before (left, *$P < 0.0001$; #$P < 0.0001$) and after (right, *$P < 0.0001$; #$P < 0.0001$; $^\&P = 0.0092$) surgery, FF-sham $n = 7$, *Agrp-Lgr4⁻/⁻*-sham $n = 7$, FF-Surgery $n = 7$, *Agrp-Lgr4⁻/⁻*-Surgery $n = 7$. (C) Tissue weights, FF-sham $n = 7$, *Agrp-Lgr4⁻/⁻*-sham $n = 7$, FF-Surgery $n = 7$, *Agrp-Lgr4⁻/⁻*-Surgery $n = 7$ (liver: *$P = 0.0347$; $P = 0.0406$.SAT: *$^\&P < 0.0001$; #$P = 0.0042$.GAT: *$^\&P < 0.0001$. BAT: $^\&P = 0.0034$.) (D) Food intake before (left, *$P = 0.0082$) and after (right, *$P = 0.0002$; $^\&P = 0.0231$), FF-sham $n = 7$, *Agrp-Lgr4⁻/⁻*-sham $n = 7$, FF-Surgery $n = 7$, *Agrp-Lgr4⁻/⁻*-Surgery $n = 7$. (E–H) TSE phenotype data, mice were placed in TSE chamber for 7 days, data for the last 3 days were recorded and analyzed, FF-sham $n = 4$, *Agrp-Lgr4⁻/⁻*-sham $n = 4$, FF-Surgery $n = 4$, *Agrp-Lgr4⁻/⁻*-Surgery $n = 4$. (E) $O_2$ consumption (dark time: *, $^\&$ and #$P < 0.0001$. Light time: *, $^\&P < 0.0001$; #$P = 0.0085$). (F) $CO_2$ production (dark time: *, #$P < 0.0001$; $^\&P = 0.0003$. Light time: *$P < 0.0001$; #$P = 0.0002$; $^\&P = 0.0437$). (G) Respiratory exchange ratio (RER). (H) Locomotor activity ($^\$P = 0.0028$; *$P = 0.0356$; $^\&P = 0.0002$). (I) Energy expenditure (dark time: *, $^\&$ and #$P < 0.0001$. Light time: *, $^\&P < 0.0001$; #$P = 0.0077$). (J, K) IPGTT (J, *$P = 0.0346$; $^\&P = 0.0113$) and IPITT (K, *$P = 0.0495$; $^\&P = 0.0498$), FF-sham $n = 7$, *Agrp-Lgr4⁻/⁻*-sham $n = 7$, FF-Surgery $n = 7$, *Agrp-Lgr4⁻/⁻*-Surgery $n = 7$. (L) Plasma insulin levels, FF-sham $n = 5$, *Agrp-Lgr4⁻/⁻*-sham $n = 5$, FF-Surgery $n = 5$, *Agrp-Lgr4⁻/⁻*-Surgery $n = 5$ (*$P = 0.0007$; $^\&P = 0.0474$). (M) Representative H&E staining of BAT (scale bar: 50 μm) sections. (N) BAT extracts were immunoblotted with indicated antibodies: anti-Adrb3 (*$P = 0.0001$; $^\&P < 0.0001$), HSL, p-HSL (*$P = 0.0028$; $^\&P = 0.0158$), UCP-1 (*$P = 0.0023$; $^\&P < 0.0001$) and β-actin, FF-sham $n = 4$, *Agrp-Lgr4⁻/⁻*-sham $n = 4$, FF-Surgery $n = 4$, *Agrp-Lgr4⁻/⁻*-Surgery $n = 4$. (O) Expression of genes related to thermogenesis and browning in BAT, FF-sham $n = 4$, *Agrp-Lgr4⁻/⁻*-sham $n = 4$, FF-Surgery $n = 4$, *Agrp-Lgr4⁻/⁻*-Surgery $n = 4$ (Adrb3: *$P < 0.0001$; Cidea: *, $^\&P < 0.0001$; Pgc-1α: $^\&P < 0.0001$; Pgc-1β: *, $^\&P < 0.0001$; Ucp-1: *, $^\&P < 0.0001$). Source data are available online for this figure.

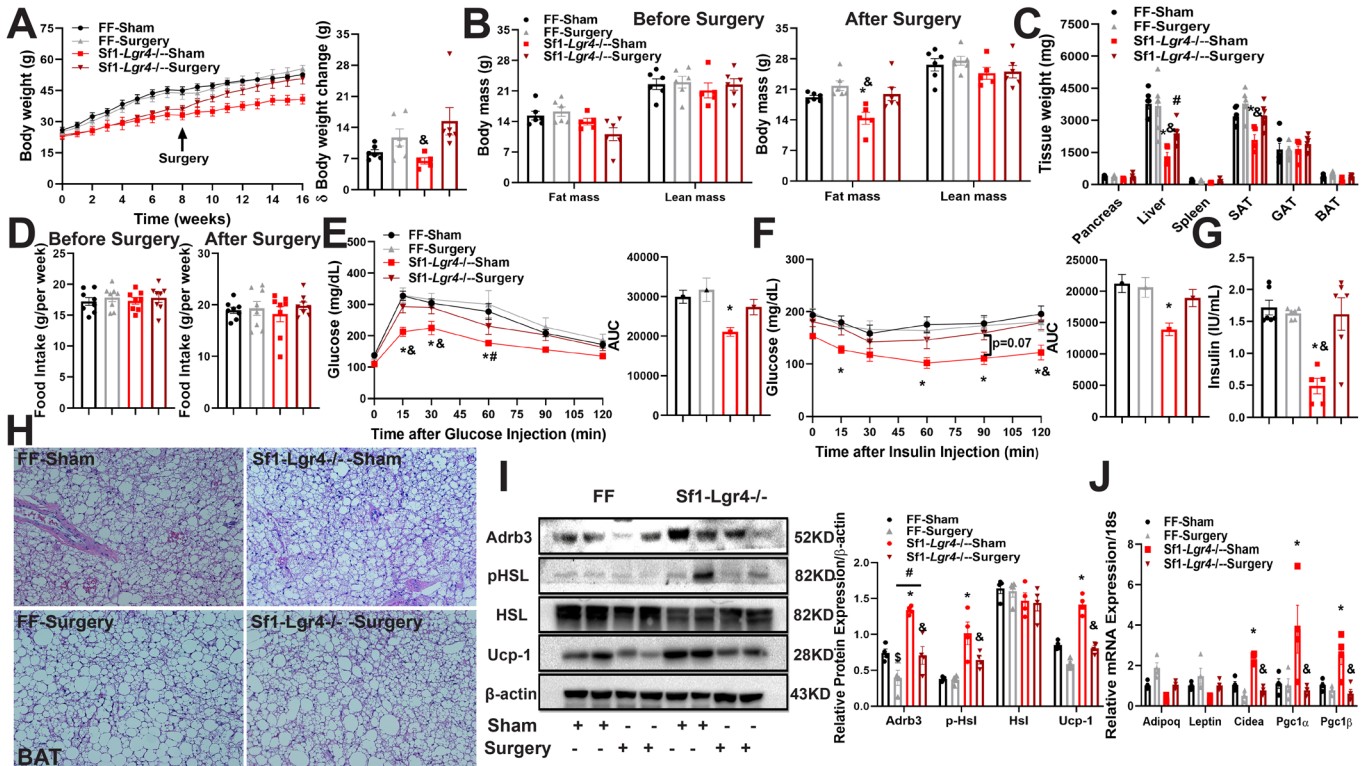

**Figure 6. Intrascapular BAT bilateral sympathectomy reverses the lean phenotype of LGR4 deficiency in *Sf1* neurons.**

Intrascapular BAT Bilateral Sympathectomy were performed on male mice fed on HFD for 8 weeks, beginning at 6 weeks. All data were shown as mean ± SEM, comparisons between more than two groups or variables were analyzed by one-way or two-way ANOVA and/or Tukey's post hoc test, $P < 0.05$, *FF-sham vs *Sf1-Lgr4$^{-/-}$*-sham, $^{\$}$FF-sham vs FF-Surgery, $^{\&}$*Sf1-Lgr4$^{-/-}$*-sham vs *Sf1-Lgr4$^{-/-}$*-Surgery, $^{\#}$ FF- Surgery vs *Sf1-Lgr4$^{-/-}$*-Surgery. (A) Body weight curve (left, *$P < 0.05$ for week 5–16; $^{\#}P < 0.05$ for week 5–8; $^{\&}P < 0.05$ for week 12–16), body weight change after surgery (right, $^{\&}P = 0.0253$), FF-Sham $n = 6$, FF-Surgery $n = 6$, *Sf1-Lgr4$^{-/-}$*-Sham $n = 5$, *Sf1-Lgr4$^{-/-}$*-Surgery $n = 6$. (B) Body composition before (left) and after (right, *$P = 0.05$; $^{\&}P = 0.0201$), FF-Sham $n = 6$, FF-Surgery $n = 6$, *Sf1-Lgr4$^{-/-}$*-Sham $n = 5$, *Sf1-Lgr4$^{-/-}$*-Surgery $n = 6$. (C) Tissue weights, FF-Sham $n = 6$, FF-Surgery $n = 6$, *Sf1-Lgr4$^{-/-}$*-Sham $n = 5$, *Sf1-Lgr4$^{-/-}$*-Surgery $n = 6$ (liver: *$P < 0.0001$; $^{\&}P = 0.0005$; $^{\#}P < 0.0001$. SAT: *$P = 0.0004$; $^{\&}P = 0.0002$). (D) Food intake before (left) and after (right), FF-Sham $n = 6$, FF-Surgery $n = 6$, *Sf1-Lgr4$^{-/-}$*-Sham $n = 5$, *Sf1-Lgr4$^{-/-}$*-Surgery $n = 6$. (E, F) IPGTT (E, 15 min: *$P = 0.0004$; $^{\&}P = 0.0247$; 30 min: *$P = 0.0257$; 60 min: *$P = 0.0025$; $^{\#}P = 0.0466$. AUC: *$P = 0.0422$) and IPITT (F), 15 min: *$P = 0.0431$; 60 min: *$P = 0.0016$; 90 min: *$P = 0.0046$;120 min: *$P = 0.0015$; AUC: *$P = 0.0215$), FF-Sham $n = 6$, FF-Surgery $n = 6$, *Sf1-Lgr4$^{-/-}$*-Sham $n = 5$, *Sf1-Lgr4$^{-/-}$*-Surgery $n = 6$. (G) Plasma insulin levels, FF-Sham $n = 6$, FF-Surgery $n = 6$, *Sf1-Lgr4$^{-/-}$*-Sham $n = 5$, *Sf1-Lgr4$^{-/-}$*-Surgery $n = 6$ (*$P = 0.0004$; $^{\&}P = 0.0037$). (H) Representative H&E staining of BAT (scale bar: 500 pixel) sections. (I) BAT extracts were immunoblotted with indicated antibodies: anti-Adrb3 ($^{\$}P = 0.0187$; *$P < 0.0001$; $^{\&}P = 0.0404$; $^{\#}P < 0.0001$), HSL, p-HSL (*$P < 0.0001$; $^{\&}P = 0.0103$), UCP-1 (*$^{,\&}P < 0.0001$) and β-actin, FF-Sham $n = 4$, FF-Surgery $n = 4$, *Sf1-Lgr4$^{-/-}$*-Sham $n = 4$, *Sf1-Lgr4$^{-/-}$*-Surgery $n = 4$. (J) Expression of genes related to thermogenesis and browning in BAT, FF-Sham $n = 4$, FF-Surgery $n = 4$, *Sf1-Lgr4$^{-/-}$*-Sham $n = 4$, *Sf1-Lgr4$^{-/-}$*-Surgery $n = 4$ (Cidea: *$P = 0.0008$, $^{\&}P < 0.0001$; Pgc-1α: *$^{,\&}P < 0.0001$; Pgc-1β: *$^{,\&}P < 0.0001$; Ucp-1: *$P = 0.0010$; $^{\&}P = 0.0002$). Source data are available online for this figure.

*Pgc-1β* were lower in both BAT and SAT, while adipocyte genes like *Adipoq* and *leptin* in SAT were higher in *Sf1-Lgr4$^{-/-}$* mice with intrascapular BAT bilateral sympathectomy relative to the *Sf1-Lgr4$^{-/-}$* transgene with sham surgery (Figs. 6J and EV4O). Taken together, these results demonstrate that intrascapular BAT bilateral sympathectomy reverses the metabolic benefits of LGR4 deficiency in Sf1 neurons.

## LGR4 deficiency in neurons increases hypothalamic leptin sensitivity

Since leptin signaling is critical for the control of food intake and energy expenditure, we next examined whether leptin signaling contributes to the metabolic benefits of neuronal LGR4 deficiency. Plasma levels of leptin in *Nestin-Lgr4$^{-/-}$*, *Sp1-Lgr4$^{-/-}$*, *Agrp-Lgr4$^{-/-}$* and *Sf1-Lgr4$^{-/-}$* mice fed HFD for 16 weeks were significantly

lower relative to sex and age matched FF mice (Figs. 7A and EV1O; Appendix Fig. S2P; Fig. EV3U). Further, reduction in plasma leptin levels was blocked by intrascapular BAT bilateral sympathectomy on *Agrp-Lgr4$^{-/-}$* or *Sf1-Lgr4$^{-/-}$* mice (Fig. EV4L,P). Hypothalamic levels of p-STAT3 protein were significantly increased, whereas suppressor of cytokine signaling 3 (SOCS-3), a suppressor of leptin sensitivity, were dramatically decreased (Fig. 7B,D). These observations indicate an increase in leptin sensitivity in *Nestin-Lgr4$^{-/-}$*, *Sp1-Lgr4$^{-/-}$*, *Agrp-Lgr4$^{-/-}$* and *Sf1-Lgr4$^{-/-}$* mice fed HFD. Further, hypothalamic expression of tyrosine hydroxylase (TH) in these mice fed HFD were higher than FF mice (Fig. 7C,D). To further assess leptin sensitivity, we treated *Nestin-Lgr4$^{-/-}$* and *Agrp-Lgr4$^{-/-}$* mice fed HFD for 8 weeks with leptin (1 mg/kg body weight/day) intraperitoneally for 3 days. Relative to FF mice, leptin induced a greater reduction in body weight and food intake in *Nestin-Lgr4$^{-/-}$* and *Agrp-Lgr4$^{-/-}$* mice (Fig. 7H,I). Further,

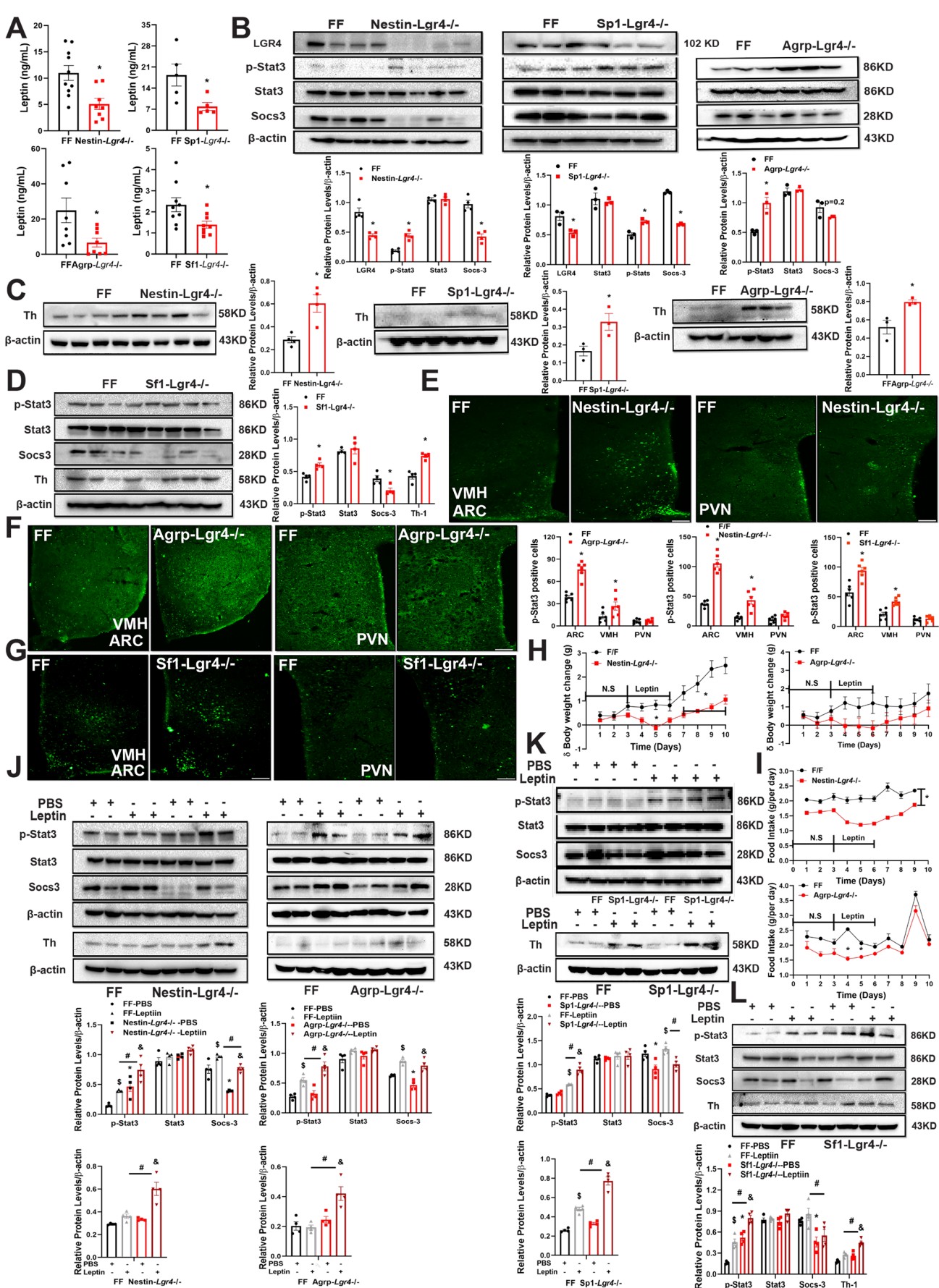

◀  **Figure 7.  LGR4 deficiency in neurons increases hypothalamic leptin sensitivity.**

Male mice were fed on HFD, beginning at 6 weeks old for 16 (**A–D**) or 8 (**F–L**) weeks. All data were shown as mean ± SEM, differences between two groups were analyzed by two-tailed Student's *t* test. Comparisons between more than two groups or variables were analyzed by one-way or two-way ANOVA and/or Tukey's post hoc test, *P* < 0.05. *FF vs *Nestin* or *Sp1* or *AgRP* or *Sf1-Lgr4*$^{-/-}$, $^{\$}$FF-PBS vs FF-Leptin, $^{\&}$*Nestin* or *Sp1*, *AgRP* or *Sf1-Lgr4*$^{-/-}$-PBS vs *Nestin* or *Sp1*, *AgRP* or *Sf1-Lgr4*$^{-/-}$-Leptin, $^{\#}$FF-Leptin vs *Nestin* or *Sp1* or *Agrp* or *Sf1-Lgr4*$^{-/-}$-Leptin. (**A**) Plasma leptin levels, (upper left, FF *n* = 10, *Nestin-Lgr4*$^{-/-}$ *n* = 8, *P* = 0.0055), (upper right, FF *n* = 5, *Sp1-Lgr4*$^{-/-}$ *n* = 5, *P* = 0.0273), (lower left, FF *n* = 8, *Agrp-Lgr4*$^{-/-}$ *n* = 8, *P* = 0.0268), (lower right, FF *n* = 9, *Sf1-Lgr4*$^{-/-}$ *n* = 9, *P* = 0.0215). (**B–D**) Hypothalamic extracts were immunoblotted with indicated leptin signaling related antibodies. anti-LGR4, p-STAT3, STAT3, Th and β-actin. (**B**) Hypothalamic extracts were immunoblotted indicated antibodies: anti-LGR4, p-STAT3, STAT3, SOCS-3, and β-actin. Left FF and *Nestin-Lgr4*$^{-/-}$ (*n* = 4, LGR4: *P* < 0.0001; p-Stat3: *P* = 0.0017; Socs3: *P* < 0.0001), Middle FF and *Sp1-Lgr4*$^{-/-}$ (*n* = 3, LGR4: *P* = 0.0034; p-Stat3: *P* = 0.0197; Socs3: *P* < 0.0001), right FF and *Agrp-Lgr4*$^{-/-}$ (*n* = 3, p-Stat3: *P* = 0.0003). (**C**) Hypothalamic extracts were immunoblotted indicated antibodies: Th and β-actin. Left, FF and *Nestin-Lgr4*$^{-/-}$ (*n* = 4, *P* = 0.0078); Middle, FF and *Sp1-Lgr4*$^{-/-}$ (*n* = 3, *P* = 0.0392); right, FF and *Agrp-Lgr4*$^{-/-}$ (*n* = 3, *P* = 0.0260). (**D**) Hypothalamic extracts were immunoblotted indicated antibodies: anti-LGR4, p-STAT3, STAT3, SOCS-3, TH-1 and β-actin, FF and *Sf1-Lgr4*$^{-/-}$ (*n* = 4, p-STAT3: *P* = 0.0344; SOCS-3: *P* = 0.0316; TH-1: *P* = 0.0002). (**E–G**) Representative hypothalamic sections stained with anti-p-STAT3 antibody. Scale bar: 100 μm. (**E**) FF and *Nestin-Lgr4*$^{-/-}$ (*n* = 3, ARC: *P* < 0.0001; VMH: *P* < 0.0001). (**F**) FF and *Agrp-Lgr4*$^{-/-}$ (*n* = 3, ARC: *P* < 0.0001; VMH: *P* = 0.0175). (**G**) FF and *Sf1-Lgr4*$^{-/-}$ (*n* = 3, ARC: *P* < 0.0001; VMH: *P* = 0.0021). (**H**) Body weight changes of mice received leptin by i.p. after fed on HFD for 8 weeks, left, FF and *Nestin-Lgr4*$^{-/-}$ (*n* = 5, Day 5: *P* = 0.0081; Day 7: *P* = 0.0160; Day 8: *P* = 0.0009; Days 9–10: *P* < 0.0001) right FF and *Agrp-Lgr4*$^{-/-}$ (*n* = 5). (**I**) Daily food intake of mice received leptin by i.p. after fed on HFD for 8 weeks. Upper, FF and *Nestin-Lgr4*$^{-/-}$ (*n* = 3, *P* < 0.05 for all days), Lower FF and *Agrp-Lgr4*$^{-/-}$ (*n* = 3, Day 4: *P* < 0.0001; Day 5: *P* = 0.0265). (**J–L**) Hypothalamus extracts after i.p. injection of leptin were immunoblotted with indicated antibodies: anti-LGR4, p-STAT3, STAT3, SOCS-3, Th and β-actin. (**J**) Left, FF and *Nestin-Lgr4*$^{-/-}$ (*n* = 4, p-Stat3: $^{\$}$*P* = 0.0069; *P* = 0.0003; $^{\#}$*P* < 0.0001; $^{\&}$*P* = 0.0018. Socs3: $^{\$}$*P* = 0.0217; *P* < 0.0001; $^{\#}$*P* = 0.0274; $^{\&}$*P* < 0.0001. Th-1: $^{\#}$*P* = 0.0009; $^{\&}$*P* = 0.0003), right, FF and *Agrp-Lgr4*$^{-/-}$ (*n* = 4, p-Stat3: $^{\$}$*P* = 0.0005; *P* < 0.0001; $^{\#}$*P* = 0.0038. Socs3: $^{\$}$*P* = 0.0024; *P* < 0.0001; Th-1: $^{\#}$*P* = 0.0009; $^{\&}$*P* = 0.0066). (**K**) FF and *Sp1-Lgr4*$^{-/-}$ (*n* = 4, p-Stat3: *P* = 0.0214; $^{\&}$*P* < 0.0001; $^{\#}$*P* = 0.0013. Socs3: $^{\$}$*P* = 0.0006; $^{\#}$*P* = 0.0008. Th-1: $^{\$}$*P* = 0.0002; $^{\#,\&}$*P* < 0.0001.), (**L**) FF and *Sf1-Lgr4*$^{-/-}$ (*n* = 4, p-Stat3: $^{\$}$*P* = 0.0001; *,$^{\#}$*P* < 0.0001; $^{\&}$*P* = 0.0003. Socs3: *P* = 0.0001; $^{\#}$*P* < 0.0001; Th-1: $^{\#}$*P* = 0.0241; $^{\&}$*P* = 0.0479). Source data are available online for this figure.

microinjection of leptin (0.1 μg in 1 μl N.S) into the lateral ventricle of *Nestin-Lgr4*$^{-/-}$, *Agrp-Lgr4*$^{-/-}$ and *Sf1-Lgr4*$^{-/-}$ mice fed HFD for 8 weeks significantly increased levels of p-STAT3 in both ventromedial hypothalamic nucleus (VMH) and arcuate nucleus (ARC) relative to age matched FF nice (Fig. 7E–G). These animals were fasted for 16 h before leptin administration. Consistently, leptin-stimulated phosphorylation of STAT3 and TH levels in hypothalamus of *Nestin-Lgr4*$^{-/-}$, *Sp1-Lgr4*$^{-/-}$, *Agrp-Lgr4*$^{-/-}$ and *Sf1-Lgr4*$^{-/-}$ mice fed HFD for 8 weeks were higher than FF mice, whereas levels of SOCS-3 in the transgenes were lower (Fig. 7J–L). Collectively, these results demonstrate that deficiency of LGR4 in neurons protects against diet-induced obesity by decreasing food intake and increasing energy expenditure through improved leptin sensitivity.

## Knockdown of neuronal LGR4 inhibits β-catenin signaling to increase leptin sensitivity

To explore the intracellular signaling pathway by which hypothalamic LGR4 regulates leptin sensitivity, we investigated hypothalamic β-catenin signaling. Knockdown of LGR4 in both of AgRP and Sp1 neurons significantly decreased β-catenin protein levels while increased phospho-β-catenin protein levels (Fig. 8A,B). This observation suggests a mechanism involving β-catenin signaling. To test this concept, we used cultured N2a cells, mouse neuroblasts with neuronal and amoeboid stem cell morphology derived from brain tissue. As shown in Fig. 8C, knock-down of LGR4 by *Lgr4* siRNA (si*Lgr4*) significantly decreased β-catenin, while increasing phospho-β-catenin. This change was associated with a decrease in SOCS-3 protein levels, and an increment in phospho-STAT 3 levels (Fig. 8C). Activation of LGR4 by Rspondin 3 significantly increased β-catenin and decreased phospho-β-catenin. These alterations were associated with increases in SOCS-3 protein levels and decrements in phospho-STAT 3 protein levels (Fig. 8D). Furthermore, the effects of Rspondin3 on leptin signaling was significantly attenuated by Dkk1, an inhibitor of β-catenin signaling (Fig. 8D).

## Discussion

The metabolic benefits of global LGR4 deficiency have been proposed to occur via activation of white adipose tissue browning (Wang et al, 2013). However, it remains unclear whether this is the direct effect of LGR4 on adipose tissue or indirect effect from LGR4 in other tissues. The presence of LGR4 receptor in hypothalamic neurons suggests an alternative mechanism may exist involving neuronal circuits. This concept is supported by the present study using a series of transgenes with neuron-specific deletion of LGR4. First, deficiency of LGR4 in nestin neurons (*Nestin-Lgr4*$^{-/-}$ mice) caused metabolic alterations characterized by reduction of food intake, body weight, adiposity and altered glucose tolerance, as well as incremental energy expenditure. Importantly, the numbers of hypothalamic neurons are unaltered in the transgene (Appendix Fig. S3A), indicating that the metabolic effects of LGR4 deficiency in neurons are likely attributable to its effect on the neuronal function rather than altered development. Second, LGR4 deficiency in Sp1 neurons (*Sp1-Lgr4*$^{-/-}$ mice) demonstrates metabolic alterations similar to *Nestin-Lgr4*$^{-/-}$ mice. This observation further supports the concept that the metabolic effects of LGR4 deficiency reside in mature neurons. It is worth of noting that the metabolic effects of LGR4 deficiency in neurons are not limited to increments in energy expenditure as previously proposed (Wang et al, 2013), but are also attributable to suppression of food intake. Third, specific deletion of LGR4 in AgRP neurons causes suppression of food intake, as well as stimulation of energy expenditure. Together with a previous report showing the presence of *Lgr4* mRNA in AGRP neurons (Wang et al, 2013), our data indicates that LGR4 signaling in AgRP neurons of ARC nuclei plays an essential role in the control of energy homeostasis by actions on suppression of food intake and stimulation of energy expenditure. Fourth, deficiency of LGR4 in Sf-1 neurons increases energy expenditure without altering food intake. Taken together, our study reveals the neuronal circuits in the hypothalamus mediating the metabolic benefits of LGR4 deficiency. LGR4 signaling in AgRP neurons of ARC nuclei functions to stimulate food intake, as well as to suppress energy expenditure. In Sf1 neurons of VMH, LGR4 signaling suppresses energy expenditure without altering food intake. Thus, the metabolic benefits of LGR4 deficiency in hypothalamus may be more

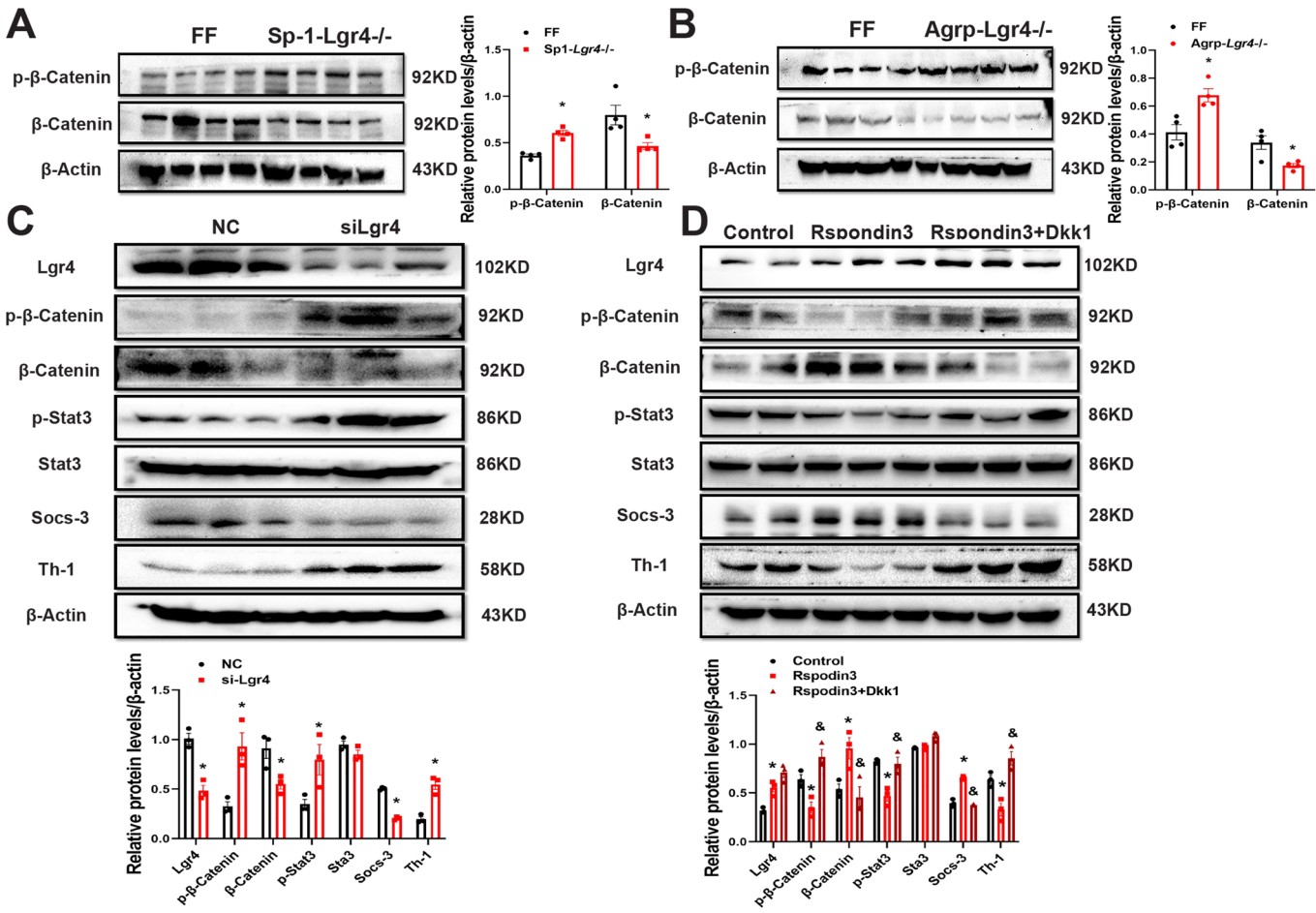

**Figure 8. Knockdown of neuronal LGR4 inhibits β-catenin to improve leptin sensitivity.**

All data were shown as mean±SEM, differences between two groups were analyzed by two-tailed Student's $t$ test. Comparisons between more than two groups or variables were analyzed by one-way or two-way ANOVA and/or Tukey's post hoc test, *Control vs Rspondin3, [&]Rspondin3 vs Rspondin3+Dkk1, $P < 0.05$. (A, B) Hypothalamus extracts from $Sp1\text{-}Lgr4^{-/-}$ (A), p-β-catenin: $P = 0.0252$; β-catenin: $P = 0.0032$) and $Agrp\text{-}Lgr4^{-/-}$ (B), p-β-catenin: $P = 0.0023$; β-catenin: $P = 0.0429$) transgenic mice were immunoblotted with indicated antibodies: p-β-catenin and β-catenin, $n = 4$. (C) Extracts of cultured neuron cells treated with $siLgr4$ were immunoblotted with indicated antibodies: p-β-catenin, β-catenin, p-STAT3, STAT3, Th-1 and SOCS-3, $n = 3$ (Lgr4: $P = 0.0001$; p-β-catenin: $P < 0.0001$; β-catenin: $P = 0.0106$; p-Stat3: $P = 0.001$; Socs-3: $P = 0.0457$; Th-1: $P = 0.0130$). (D) Extracts of cultured neuron cells treated with Rspondin 3 (100 ng/mL) and Dkk1 (50 ng/mL) were immunoblotted with indicated antibodies: p-β-catenin, β-catenin, p-STAT 3, STAT 3, Th-1 and SOCS-3, $n = 3$ (Lgr4: *$P = 0.0117$; p-β-catenin: *$P = 0.0015$, [&]$P < 0.0001$; β-catenin: *,[&]$P < 0.0001$; p-Stat3: *$P = 0.001$, [&]$P = 0.0003$; Socs-3: *$P = 0.0052$, [&]$P = 0.0027$; Th-1: *$P = 0.0007$, [&]$P < 0.0001$). Source data are available online for this figure.

profound, including reduction of food intake and increased energy expenditure. Targeting hypothalamic LGR4 may thus provide a more promising strategy for the intervention of obesity and its related metabolic dysfunction.

The intracellular signaling pathway by which hypothalamic LGR4 regulates food intake and energy expenditure remains unknown. The present study indicates that deficiency of LGR4 in hypothalamic neurons reverses the downregulation of hypothalamic leptin signaling induced by HFD, causing an improvement of leptin sensitivity. Levels of SOCS-3, an inhibitor of leptin signaling, are decreased in all four lines of LGR4 deficient transgenes, including $nestin\text{-}Lgr4^{-/-}$, $Sp1\text{-}Lgr4^{-/-}$, $Agrp\text{-}Lgr4^{-/-}$, and $Sf1\text{-}Lgr4^{-/-}$ mice. Phosphorylation of STAT3 was observed to be increased. These results indicate that LGR4 in neurons may function to control food intake and energy expenditure by regulating SOCS-3-STAT3 signaling. How LGR4 in hypothalamic neurons alters

SOCS-3-STAT3 signaling remains unclear. Because we observed a significant decrement of β-catenin in the hypothalamus derived from LGR4 deficient mice (Fig. 8), we propose that β-catenin signaling may mediate the effect of LGR4 on SOCS3-STAT3 signaling in hypothalamic neurons. This concept is supported by the observation that inhibition of β-catenin blocks the effects of activation of LGR4 by R-spondin 3 on the phosphorylation of STAT3 in cultured neurons. Previous studies have shown that β-catenin signaling may increase levels of phosphorylated STAT3 in macrophages (Wang et al, 2018).

Leptin signaling in the hypothalamus is critical for the control of food intake and energy expenditure. In particular, leptin signaling in ARC and VMH increases sympathetic output and subsequent lipolysis in brown and beige adipose tissues. In this study, administration of exogenous leptin induced a significant reduction of food intake and adiposity in HFD-fed mice with neuronal deficiency of LGR4. Further,

deficiency of LGR4 in AgRP and Sf1 neurons significantly increases levels of hypothalamic tyrosine hydroxylase (TH), indicating an increase in the sympathetic activity. Consistently, intrascapular BAT bilateral sympathectomy reverses the lean phenotype and metabolic consequences associated with LGR4 deficiency in AgRP or Sf1 neurons, ranging from diet-induced adiposity, liver steatosis, to glucose and insulin tolerance. Thus, we have defined the leptin-SNA-BAT pathway as the potential mechanism by which hypothalamic LGR4 signaling controls food intake and energy expenditure.

In our hands, female Sp1-Lgr4$^{-/-}$ and AgRP-Lgr4$^{-/-}$ mice showed a milder phenotype than male mice. This observation is consistent with other reports showing that male and female C57BL/6J mice exhibit strikingly different responses in weight, food consumption, locomotor activity, energy expenditure, etc. (Casimiro et al, 2021; Sims et al, 2013). Mechanism underlying the distinct responses between male and female mice may be attributed to the hormonal physiological differences. A series of evidence has suggested that estrogen and G-protein coupled estrogen receptor may have a direct effect on the metabolic physiology including food intake (Davis et al, 2014; Musatov et al, 2007; Xu et al, 2011). Other mechanism such as chromosomal genetic differences (Butera, 2010; Geer and Shen, 2009; Kirsch et al, 2003; Meyer et al, 2011), may also contribute to the sex difference in metabolic phenotype.

In summary, our studies demonstrate that hypothalamic LGR4 signaling is crucial for control of food intake and energy expenditure in mice. Activation of LGR4 in hypothalamic neurons results in energy conservation by increasing food intake and concurrently inhibiting energy expenditure via the suppression of leptin-SNA-BAT axis in a manner dependent on β-catenin. Antagonism of hypothalamic LGR4 may thus provide a novel strategy for the intervention of obesity and its related metabolic dysfunction.

# Methods

### Reagents and Tools Table

| Reagent/resource | Reference or source | Identifier or catalog number |
|---|---|---|
| **Experimental models** | | |
| *Nestin-Cre* mice | The Jackson Laboratory | IMSR_JAX:003771 |
| *Sp1-Cre* mice | The Jackson Laboratory | IMSR_JAX:003966 |
| *Sf1-Cre* mice | The Jackson Laboratory | IMSR_JAX:012462 |
| *Agrp-ERT-Cre* mice | University of Michigan | N/A |
| *Pomc-Cre* | University of Michigan | N/A |
| *Lgr4$^{flox/flox}$* mice | University of Michigan | N/A |
| Neuro-2a cell line | ATCC | CCL-131 ™ |
| **Antibodies** | | |
| LGR4 | Invitrogen | PA5-109908 |
| Phospho--Catenin (Ser675) (D2F1) XP Rabbit mAb | CST | 4176 |
| β-Catenin (D10A8) XP® Rabbit mAb | CST | 8480 |
| c-Fos (Ab-5) (4-17) Rabbit pAb | EMD Millipore | PC38 |
| Phospho-Stat3 (Tyr705) Antibody | CST | 9131 |
| Stat3 Antibody | CST | 9132 |

| Reagent/resource | Reference or source | Identifier or catalog number |
|---|---|---|
| SOCS3 Polyclonal antibody | Proteintech | 14025-1-AP |
| Tyrosine Hydroxylase Antibody | CST | 2792 |
| Anti-ADRB3 antibody | MilliporeSigma | SAB4500584 |
| HSL Antibody | CST | 4107 |
| Phospho-HSL (Ser563) Antibody | CST | 4139 |
| UCP1 (D9D6X) Rabbit mAb | CST | 14670 |
| Beta Actin Recombinant antibody | Proteintech | 81115-1-RR |
| **Oligonucleotides and other sequence-based reagents** | **Forward** | **Reverse** |
| fasn (FAS) | TGGGTTCTAGCCAGCAGAGT | ACCACCAGAGAC CGTTATGC |
| gpam (GPAT) | CACACGAGCAGGAAAGATGA | GGACTGCATAGA TGCTGCAA |
| dgat1 (DGAT1) | TTCCGCCTCTGGGCATT | AGAATCGGCCCA CAATCCA |
| srebf1 (SREBP1) | GGAGCCATGGATTGCACATT | GGAAGTCACTGT CTTGGTTGTTGA |
| apob (apoB) | TCACCATTTGCCCTCAACCTAA | GAAGGCTCTTTG GAAGTGTAAAC |
| apoe (apoE) | CCGACATGGAGGATCTACGC | TCTCCATCAGGT TTGCCCAC |
| cd36 (CD36) | TGGTCAAGCCAGCTAGAAA | CCCAGTCTCATT TAGCCAC |
| LDLR | GCATCAGCTTGGACAAGGTGT | GGGAACAGCCA CCATTGTTG |
| Pnpla2(atgl) | GACAGCTCCACCAACATCCA | GAAGGCAGATG GTCACCCAA |
| cpt1a (CPT1α) | ATCGTGGTGGTGGGTGTGATAT | ACGCCACTCAC GATGTTCTTC |
| Adrb3 | CAGGCGCCACACGAGATG | GCGGGCGATGG CTATGAT |
| Cidea | CATACATCCAGCTCGCCCTT | TGTATCGCCCA GTACTCGGA |
| Ucp-1 | GGACGACCCCTAATCTAATG | CATTAGATTAG GGGTCGTCC |
| Adiponectin | TAgAgAAgAAAgCCAgTAAATg | ACCAgTATCAgg AAAAgAATgT |
| Leptin | TAGCCAATGACCTGGAGAATC | TCAGCATTCAG GGCTAACATC |
| Ppargc1a(Pgc-1α) | GGCCTAACTCCTCCCACAAC | AGGGATGACC GAAGTGCTTG |
| Ppargc1b(Pgc-1β) | CCCTCGATGTGCCTGGATAC | CAAAGGAGCAG GAGAAGGGG |
| 18 s | CGATGCTCTTAGCTGAGTGT | GGTCCAAGAAT TTCACCTCT |
| siLGR4 | GUAUCGUGCAAACACUUAAUU | UUAAGUGUUUG CACGAUACUG |
| **Chemicals, enzymes and other reagents** | | |
| Mouse Ins1/Insulin ELISA Kit | MilliporeSigma | RAB0817 |
| Mouse Leptin ELISA Kit | MilliporeSigma | RAB0334 |
| D-(+)-Glucose | MilliporeSigma | G7021 |
| Recombinant-human-insulin | Thermo Fisher Scientific | A11382II |

| Reagent/resource | Reference or source | Identifier or catalog number |
|---|---|---|
| Recombinant Mouse Leptin | R&D | 498-OB |
| Triglyceride Colorimetric Assay Kit | Cayman Chemical | 10010303 |
| Cholesterol Fluorometric Assay Kit | Cayman Chemical | 10007640 |
| RNAiMAX Transfection Reagent | Fisher Scientific | 13778500 |
| Recombinant Mouse R-Spondin 3 Protein | R&D | 4120-RS |
| Recombinant Mouse Dkk-1 Protein | R&D | 5897-DK |
| Gibco™ DMEM | Fisher Scientific | 11995073 |
| Fetal Bovine Serum | Fisher Scientific | 16140071 |
| **Software** | | |
| GraphPad Prism 8 | https://www.graphpad.com/ | |
| ImageJ | https://imagej.net/ | |
| **Other** | | |
| Bio-Rad ChemiDoc XRS + Gel Imaging System | https://www.bio-rad.com/ | |
| ABI QuantStudio 3 qPCR system | https://www.thermofisher.com/ | |
| Olympus BX53 Microscope | https://www.olympus-ims.com/ | |
| Leica CM1850 Cryostat | https://www.leicabiosystems.com/ | |

## Animals and treatments

### Animals

Animals were housed in a temperature-controlled environment with 12-h:12-h light-dark cycles, and access to food (Normal Chow Diet (NCD) or 60% High Fat Diet (HFD); Research Diet) and water ad libitum. All animal experiments were approved and performed according to the Institutional Animal Care and Use Committee (IACUC) of the University of Michigan in accordance with the National Institutes of Health (NIH) guidelines. Nestin-Cre (RRID: IMSR_JAX:003771 C57BL/6J background), Sp1-Cre (RRID: IMSR_JAX:003966 C57BL/6J background), Sf1-Cre mice (RRID: IMSR_JAX:012462) were purchased from Jackson Laboratory (Bar Harbor, ME). Agrp-ERT-Cre mice were donated by Dr. Martin G. Myers, Jr. from the University of Michigan. Pomc-Cre (C57BL/6J background) mice were donated by Dr. Langyou Rui from the University of Michigan. Agrp-ERT-Cre, Sf1-Cre were cross-bred to C57BL/6 background for at least eight generations. Nestin-Lgr4$^{-/-}$, Sp1-Lgr4$^{-/-}$, Agrp-Lgr4$^{-/-}$, Pomc-Lgr4$^{-/-}$, Sf1-Lgr4$^{-/-}$ mice were generated by cross-breeding Nestin-Cre, Sp1-Cre, Agrp-ERT-Cre, Pomc-Cre, or Sf1-Cre mice with Lgr4$^{flox/flox}$ mice respectively. Cre positive and Lgr4$^{flox/flox}$ homozygous mice were used as the knockout group, whereas littermate mice with Cre negative and Lgr4$^{flox/flox}$ homozygous mice (FF) were used as the control group throughout the experiments. Deletion of Lgr4 was confirmed by genotyping and western blot. To induce obesity, 6 weeks old mice were fed with 60% high fat diet (HFD) for 8 or 16 weeks. Both male and female mice were used unless indicated otherwise.

### Intracerebroventricular injection

Male mice fed HFD for 8 weeks were isoflurane-anesthetized and mounted on an Ultra Precise Small Animal Stereotaxic Alignment System (David Kopf Instruments). A small opening was made in the skull, and the bregma was located. A guarded 26-gauge needle was used to create a hole 0.2 mm caudal to the bregma and 1.0 mm lateral to the midline at a depth of 2.25 mm. A 1.0 µl Hamilton syringe (Hamilton, Reno, NV) was used to inject 100 ng of murine leptin (R&D systems) or cerebrospinal fluid vehicle in a volume of 1.0 µl into the right cerebral ventricle.

### Intrascapular brown adipose tissue bilateral sympathectomy

Afferent sympathectomy of brown adipose tissue (BAT) was performed in mice fed HFD for 8 weeks as described previously(Engel et al, 1992; Niijima et al, 1984). In brief, intrascapular BAT was exposed by incising the overlaying skin and adjacent muscles. Under a surgical microscope, five sympathetic nerve bundles supplying each lobe of BAT were identified, carefully lifted with forceps, and severed. A similar procedure without nerve division was performed in sham-operated groups. The mice were placed on heating pads during surgery and until recovery.

### NMR body composition measurement

Body composition was measured using an EchoMRI (Echo Medical Systems, USA). Mice were placed in the clear designed cylindrical tube. The tube is placed within the machine and each mouse was scanned for 1–2 min.

### IPGTT and ITT

For glucose tolerance testing (GTT), mice were fasted for 16 h and injected with D-glucose (1.5 g/Kg, Sigma-Aldrich, USA) intraperitoneally. For insulin tolerance testing (ITT), mice were fasted for 6 h and intraperitoneally injected with recombinant human insulin (Thermo Fisher Scientific, USA) at the dose of 0.5 or 0.75 U/Kg for mice fed NCD or HFD, respectively. Tail vein blood samples were collected at the indicated time points and glucose concentration measured using a Biosen glucose and lactate analyzer (BiosenC-Line, EKF, Germany).

### Energy expenditure

Indirect gas calorimetry, oxygen consumption ($VO_2$) and carbon dioxide production ($VCO_2$) were used to calculate various metabolic parameters, including the respiratory exchange rate (RER) and substrate utilization, food and liquid intake. Locomotor activity was measured using a 24-cage TSE PhenoMaster system (TSE Systems; Germany).

### Cold exposure

Mice were placed in a 4 °C cold chamber (Thermo Fisher Scientific, USA) for 6 h and rectal temperature was measured every hour during the cold challenge with a rectal probe (Thermo Fisher Scientific, USA).

### Intraperitoneal administration of leptin

Mice were injected with leptin (1 mg/Kg body weight, R&D, USA) intraperitoneally at 6 pm daily for 3 days. Normal saline injection (NS) was used as control. Body weight and food intake were recorded daily.

### Measurement of plasma insulin and leptin

Blood samples were collected from cardiac veins and stocked in plastic blood collection tubes with K2 EDTA (BD Vacutainer™, USA). Plasma was collected by centrifuge and held at −80 °C before

use. Plasma insulin and leptin were measured using insulin and leptin ELISA kits (Sigma-Aldrich, USA), respectively.

### Triglyceride and cholesterol measurement

Plasma and liver lipid were measured using triglyceride and cholesterol kits (Cayman, USA). For liver lipid, liver tissues were homogenized for total lipid extraction as described before (Bligh and Dyer, 1959). Total lipids were re-suspended in 400 μL ethanol and incubated at 55 °C for 10 min before use.

## Cell line and cell culture

Mouse neuroblast cell line Neuro-2a (N2a, CCL-131™) was obtained from American Type Culture Collection (Rockville, MD, USA). N2a cells were cultivated in Dulbecco's modified Eagle's medium (DMEM, Gibco, Rockville, MD, USA) supplemented with 10% fetal bovine serum (FBS, Invitrogen, Carlsbad, CA, USA). Cells were cultured in medium without serum for 6 h before exposure to exogenous Rspondin 3 (100 ng/mL, R&D, USA).

## Transient transfection

LGR4 siRNA (50 nM, Thermo Fisher, USA), and negative controls (siNC) were transfected into N2a cells using the Lipofectamine RNAiMAX Transfection Reagent (Invitrogen, CA, USA) according to the manufacturer's instructions. The efficiency of transfection was evaluated by western blotting. The LGR4 siRNA sequences are shown in Reagents and tools table.

## DKK1 inhibitory assay

To validate to the role of LGR4 in activation of β-catenin signaling, N2a cells pre-treated with Rspondin3 were treated with DKK1 (50 ng/mL, R&D, USA). Western blotting was used to examine protein expression of Wnt/β-catenin signaling-related molecules.

### RNA isolation and real-time PCR

Total RNA was extracted from tissues using TRIzol reagent (Invitrogen, USA) in accordance with the manufacturer's instructions. In all, 1 μg of RNA was transcribed to complementary DNA with the iScript™ Reverse Transcription Supermix (Bio-Rad, USA). Real-time PCR was carried out on the QuantStudio 3 Real-Time PCR System (Applied Biosystems, Thermo Fisher, USA) using QuantiTect Probe RT-PCR kit (Qiagen, USA). Primers used in this study are provided in Reagents and tools table. Data were normalized to 18S and analyzed using the $^{\triangle\triangle}$CT method.

### Protein preparation and western blotting

Proteins from tissues were prepared using radioimmunoprecipitation assay (RIPA) buffer. All protein samples were subjected to concentration determination and immunoblot assay with the indicated antibodies. Actin was used as the internal controls. The membranes were visualized using a ChemiDoc XRS+ Gel Imaging System (Bio-Rad, USA) in accordance with the manufacturer's guide. Detailed information on the antibodies used in this study is provided in Reagents and tools table.

### Hematoxylin-eosin staining

Mouse tissues were fixed in 4% paraformaldehyde (PFA) and embedded in paraffin. Sections of 5-μm thickness were stained with hematoxylin and eosin (Thermo Fisher Scientific, USA) according to standard protocols.

### Brain section and staining

Mice were perfused with formalin via trans-cardial perfusion (Münzberg et al, 2003). Brains were removed after 4% PFA infusion and placed in 4% PFA overnight, then transferred to 30% sucrose. Brains were embedded with OCT and sectioned into 20 μm with a cryostat (Lecia, USA). Floating staining with p-Stat3 was applied to these sections as described (Cellini and Lombardo, 2020).

### Statistics

Data arer presented as mean ± SEM. Differences between two groups were analyzed by two-tailed Student's $t$ test. Comparisons between more than two groups or variables were analyzed by one-way or two-way ANOVA and/or Tukey's post hoc test using GraphPad Prism 8. A $P$ value of less than 0.05 was considered significant.

## Data availability

This study includes no data deposited in external repositories.

The source data of this paper are collected in the following database record: biostudies:S-SCDT-10_1038-S44319-025-00398-5.

## Peer review information

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

## Acknowledgements

We would like to thank Dr.Micheal Mulholland, Dr. Randy Seeley and Stace Kernodle at the University of Michigan for technical assistance and critical reading of the manuscript. This work was supported by the NIH/NIDDK R01DK129360, R01DK112755, R01DK110273, P30DK089503.

## Author contributions

**Liping Zhang**: Conceptualization; Data curation; Formal analysis; Writing—original draft; Writing—review and editing. **Yuan Li**: Formal analysis. **Wenbin Gao**: Data curation. **Ziru Li**: Data curation. **Tong Wu**: Data curation. **Chunhui Lang**: Data curation. **Liangyou Rui**: Data curation. **Weizhen Zhang**: Conceptualization; Supervision; Funding acquisition; Writing—original draft; Writing—review and editing.

Source data underlying figure panels in this paper may have individual authorship assigned. Where available, figure panel/source data authorship is listed in the following database record: biostudies:S-SCDT-10_1038-S44319-025-00398-5.

## Disclosure and competing interests statement

The authors declare no competing interests.

# Expanded View Figures

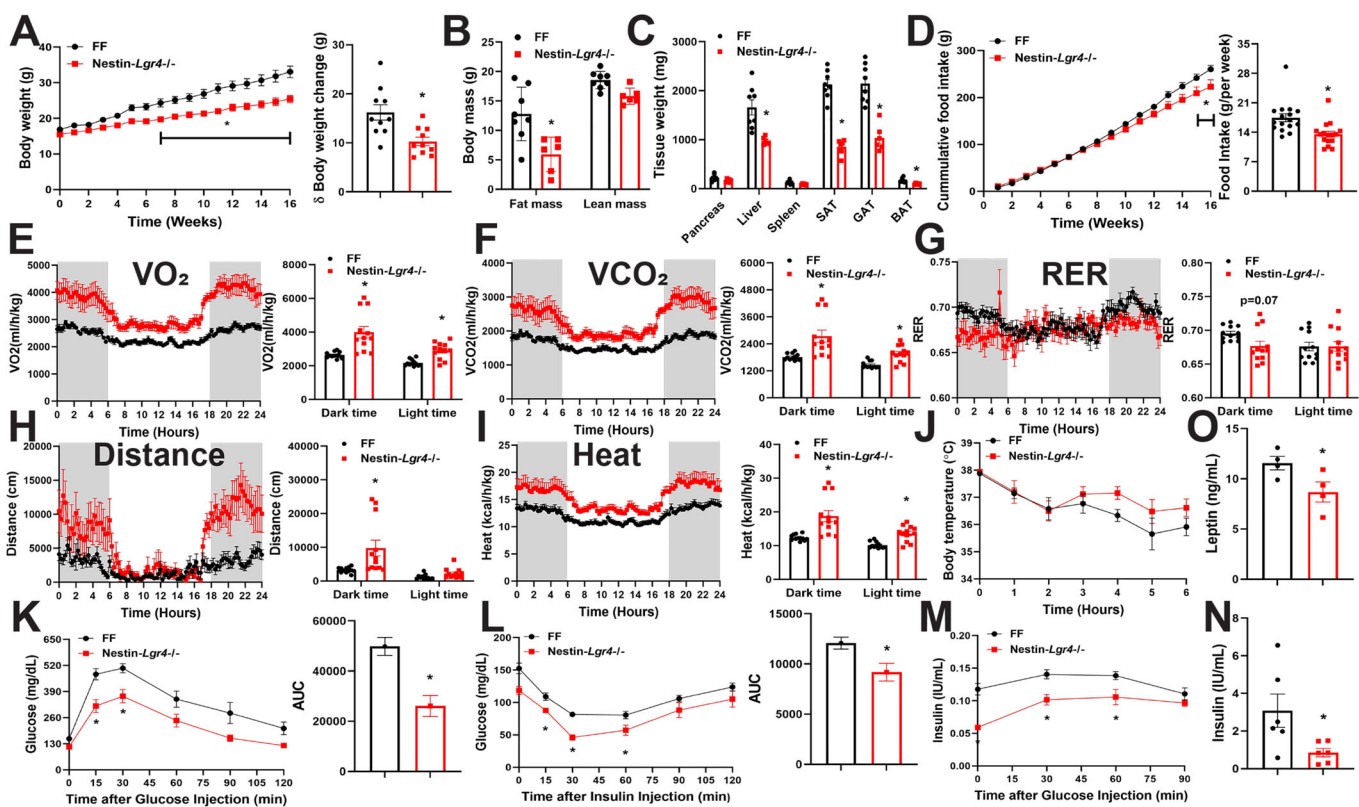

**Figure EV1. Metabolic effects of *Lgr4* deficiency in *nestin* neurons of female mice fed HFD.**

Female mice at 6 weeks old were fed HFD for 16 weeks. All data were shown as mean±SEM, differences between 2 groups were analyzed by two-tailed Student's *t* test. Comparisons between more than 2 groups or variables were analyzed by one-way or two-way ANOVA and/or Tukey's post hoc test, *$P<0.05$. **(A)** Body weight (left, $P < 0.05$ for week 7 to 16) and body weight change (right, $P = 0.0042$), FF $n = 10$, *Nestin-Lgr4*$^{-/-}$ $n = 10$. **(B)** Body mass (Fat mass: $P = 0.0005$), FF $n = 8$, *Nestin-Lgr4*$^{-/-}$ $n = 6$. **(C)** Tissue weights, FF $n = 8$, *Nestin-Lgr4*$^{-/-}$ $n = 6$ (liver: $P = 0.002698$; SAT: $P < 0.000001$; GAT: $P = 0.000063$; BAT: $P = 0.002363$). **(D)** Cumulative food intake (left, $P < 0.05$ for week 14 to 16) and weekly food intake (right, $P = 0.0024$), FF $n = 8$, *Nestin-Lgr4*$^{-/-}$ $n = 6$. **(E–I)** TSE phenotype data, mice were placed in TSE chamber for 7 days, data from last 3 days were recorded and analyzed, FF $n = 4$, *Nestin-Lgr4*$^{-/-}$ $n = 4$. **(E)** O$_2$ consumption (dark time: $P < 0.0001$; light time: $P = 0.0187$). **(F)** CO$_2$ production (dark time: $P = 0.0001$; light time: $P = 0.0446$). **(G)** Respiratory Exchange Ratio (RER). **(H)** locomotor activity (dark time: $P = 0.0009$). **(I)** Energy expenditure (dark time: $P < 0.0001$; light time: $P = 0.0218$). **(J)** Rectal body temperature under 4 °C, FF $n = 6$, *Nestin-Lgr4*$^{-/-}$ $n = 5$. **(K, L)** IPGTT (**K**, $P = 0.0269$) and IPITT (**L**, $P = 0.0155$), FF $n = 8$, *Nestin-Lgr4*$^{-/-}$ $n = 6$. **(M)** Insulin secretion after glucose injection, FF $n = 6$, *Nestin-Lgr4*$^{-/-}$ $n = 6$ (0 min: $P < 0.0001$; 30 min: $P = 0.0044$; 60 min: $P = 0.0200$). **(N)** Plasma Insulin, FF $n = 6$, *Nestin-Lgr4*$^{-/-}$ $n = 6$ ($P = 0.0351$). **(O)** Plasma Leptin, FF $n = 4$, *Nestin-Lgr4*$^{-/-}$ $n = 4$ ($P = 0.0548$) Source data are available online for this figure.

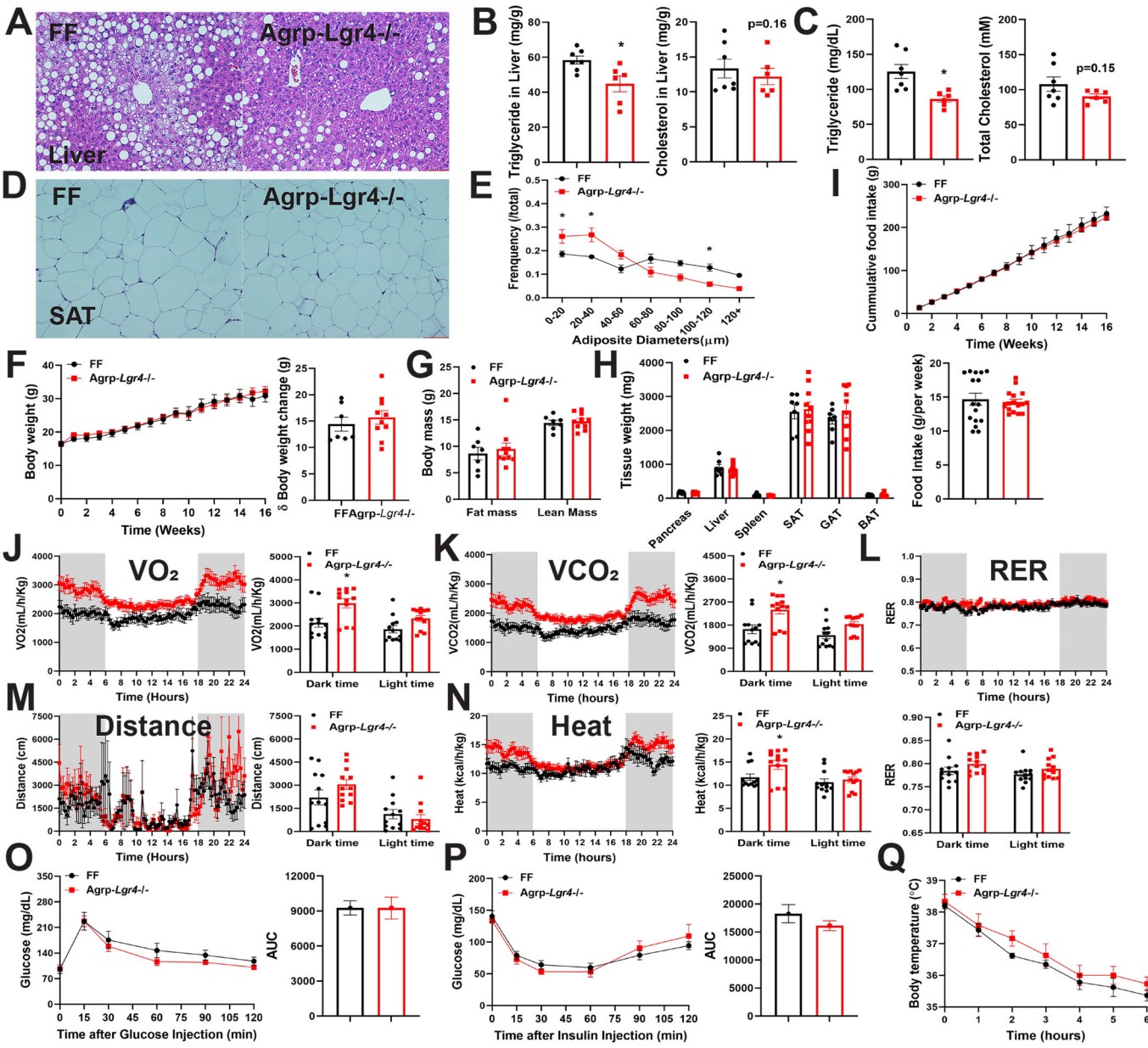

**Figure EV2. Metabolic effects of Lgr4 deficiency in AgRP neurons of mice fed HFD.**

Male (**A–E**) and female mice (**F–Q**) were fed on HFD at 6 weeks old for 16 weeks. All data were shown as mean±SEM, differences between 2 groups were analyzed by two-tailed Student's *t* test. Comparisons between more than two groups or variables were analyzed by one-way or two-way ANOVA and/or Tukey's post hoc test, *P < 0.05. (**A**) Representative H&E staining of liver (scale bar: 500 pixel) sections. (**B**) Triglyceride content (left, P = 0.0166) and total cholesterol content (right) in liver, FF n = 7, Agrp-Lgr4−/− n = 6. (**C**) Triglyceride content (left, P = 0.0059) and total cholesterol content (right) in plasma, FF n = 7, Agrp-Lgr4−/− n = 6. (**D**) Representative H&E staining of SAT (scale bar: 500 pixel) sections. (**E**) Adipocyte size of SAT, FF n = 6, Agrp-Lgr4−/− n = 6 (0–20 μm: P = 0.0177; 20–40 μm: P = 0.0020; 100–120 μm: P = 0.0250). (**F**) Body weight (left) and body weight change (right), FF n = 7, Agrp-Lgr4−/− n = 10. (**G**) Body mass, FF n = 7, Agrp-Lgr4−/− n = 10. (**H**) Tissue weights, FF n = 7, Agrp-Lgr4−/− n = 10. (**I**) Cumulative food intake (left) and weekly food intake (right), FF n = 7, Agrp-Lgr4−/− n = 10. (**J–N**) TSE phenotype data, mice were placed in TSE chamber for 7 days, data from last 3 days were recorded and analyzed, FF n = 4, Agrp-Lgr4−/− n = 4. (**J**) O2 consumption (dark time: P = 0.0024). (**K**) CO2 production (dark time: P = 0.0011). (**L**) Respiratory Exchange Ratio (RER). (**M**) Locomotor activity. (**N**) Energy expenditure (dark time: P = 0.0325). (**O, P**) IPGTT (**O**) and IPITT (**P**), FF n = 5, Agrp-Lgr4−/− n = 5. (**Q**) Rectal body temperature under 4 °C, FF n = 6, Agrp-Lgr4−/− n = 6. Source data are available online for this figure.

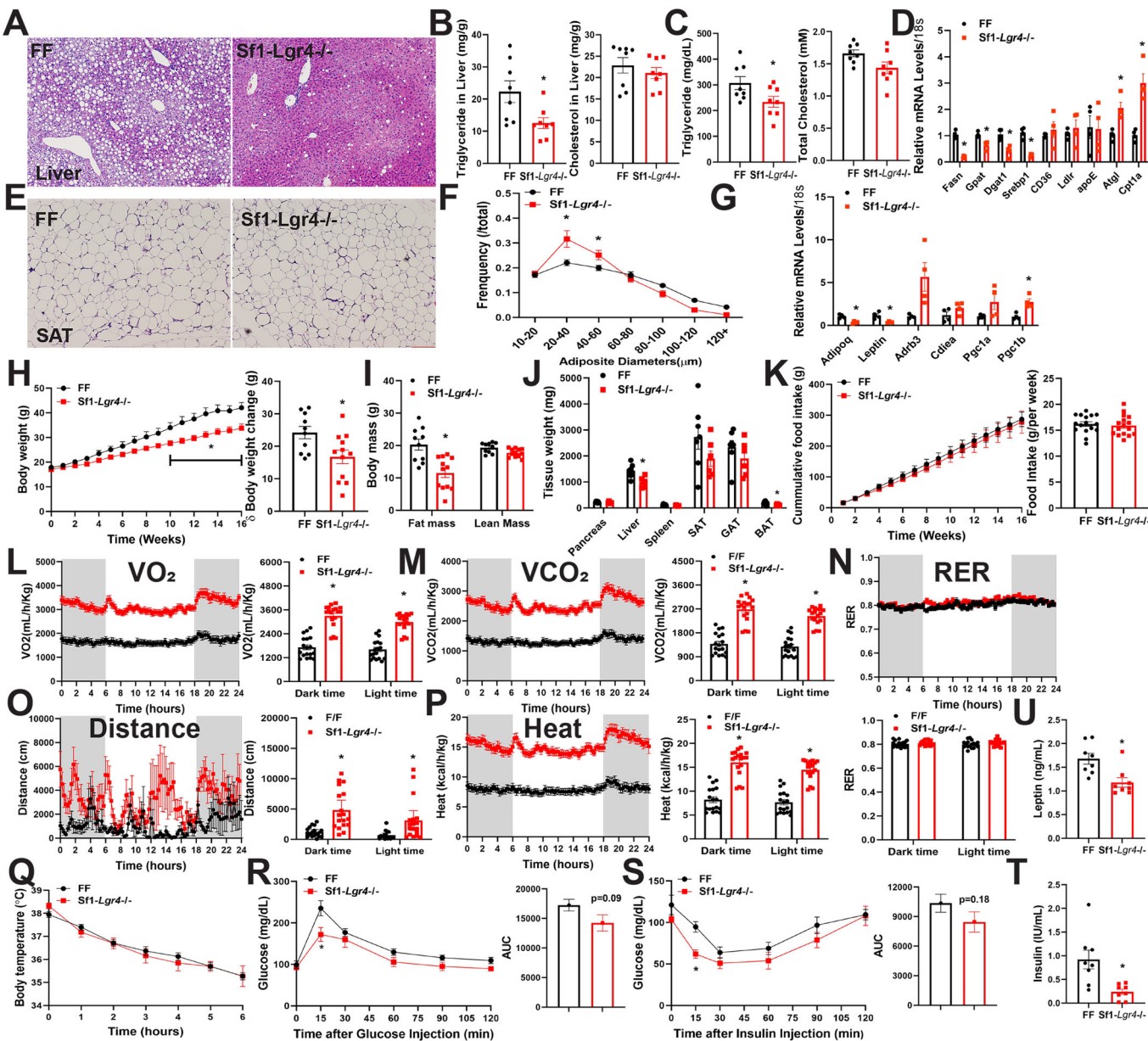

**Figure EV3. Metabolic effects of Lgr 4 knockdown in Sf1 neurons of mice fed HFD.**

Male mice (**A–G**) and female mice (**H–T**) were fed HFD at 6 weeks old for 16 weeks. All data were shown as mean±SEM, Differences between two groups were analyzed by two-tailed Student's *t* test. Comparisons between more than two groups or variables were analyzed by one-way or two-way ANOVA and/or Tukey's post hoc test, *$P < 0.05$. (**A**) Representative H&E staining of liver (scale bar: 50 μm) sections. (**B**) Triglyceride content (left, $P = 0.0216$) and total cholesterol content (right) in liver, FF $n = 8$, Sf1-Lgr4−/− $n = 8$. (**C**) Triglyceride content (left, $P = 0.0417$) and total cholesterol content (right) in plasma, FF $n = 8$, Sf1-Lgr4−/− $n = 8$. (**D**) Expression of genes related to lipid genesis, transport and β-oxidation in liver tissue, FF $n = 4$, Sf1-Lgr4−/− $n = 4$ (Fasn: $P = 0.000293$; Gpat: $P = 0.024476$; Dgat1: $P = 0.024476$; Srebp1: $P = 0.001694$; Atgl: $P= 0.024476$; Cpt1α: $P = 0.012742$). (**E**) Representative H&E staining of SAT (scale bar: 50 μm) sections. (**F**) Adipocyte size of SAT, FF $n = 6$, Sf1-Lgr4−/− $n = 5$ (20–40 μm: $P < 0.0001$; 40–60 μm: $P = 0.0499$). (**G**) Expression of genes related to lipolysis and browning in SAT, FF $n = 4$, Sf1-Lgr4−/− $n = 4$ (Adipoq: $P = 0.009246$; Leptin: $P = 0.027954$; Pgc1β: $P = 0.026012$). (**H**) Body weight (left, $P < 0.05$ for week 10 to 16) and body weight change (right, $P = 0.0173$), FF $n = 10$, Sf1-Lgr4−/− $n = 12$. (**I**) Body mass, FF $n = 10$, Sf1-Lgr4−/− $n = 12$ (Fat mass: $P < 0.0001$). (**J**) Tissue weights, FF $n = 7$, Sf1-Lgr4−/− $n = 6$ (liver: $P = 0.005946$; BAT: $P = 0.022265$). (**K**) Cumulative food intake (left) and weekly food intake (right), FF $n = 10$, Sf1-Lgr4−/− $n = 12$. (**L–P**) TSE phenotype data, mice were placed in TSE chamber for 7 days, data from last 3 days were recorded and analyzed, FF $n = 6$, Sf1-Lgr4−/− $n = 6$. (**L**) O₂ consumption (dark time: $P < 0.0001$; light time: $P < 0.0001$). (**M**) CO₂ production (dark time: $P < 0.0001$; light time: $P < 0.0001$). (**N**) Respiratory exchange ratio (RER). (**O**) Locomotor activity (dark time: $P = 0.022250$; light time: $P = 0.010912$). P Energy expenditure (dark time: $P < 0.0001$; light time: $P < 0.0001$). (**Q**) Rectal body temperature under 4 °C, FF $n = 8$, Sf1-Lgr4−/− $n = 7$. (**R–T**) IPGTT (**R**) and IPITT (**S**), FF $n = 8$, Sf1-Lgr4−/− $n = 8$. (**T**) Plasma insulin levels, FF $n = 8$, Sf1-Lgr4−/− $n = 8$ ($P = 0.005422$). (**U**) Plasma leptin levels, FF $n = 8$, Sf1-Lgr4−/− $n = 8$ ($P = 0.0058$). Source data are available online for this figure.

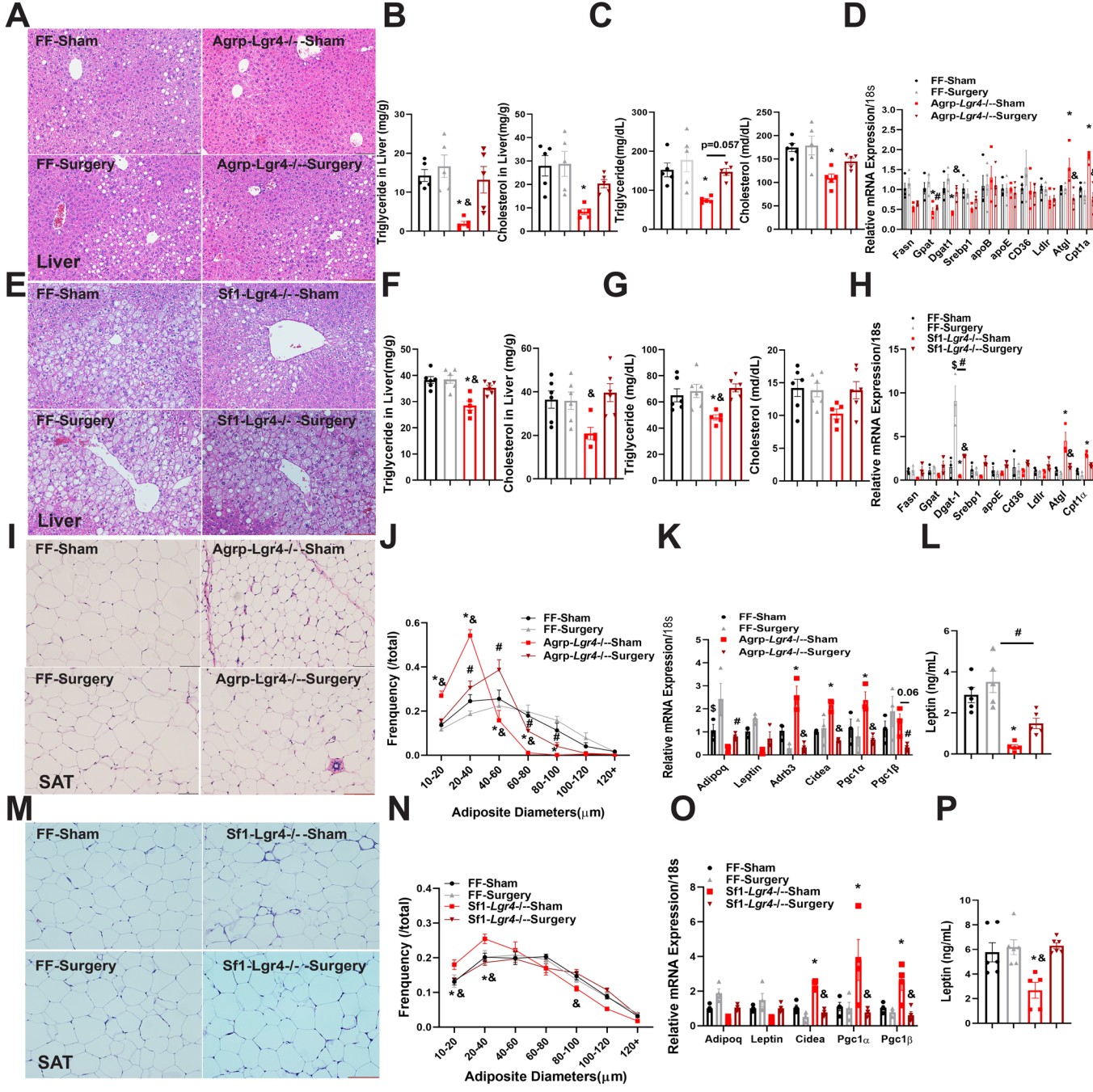

◀

**Figure EV4. Intrascapular BAT bilateral sympathectomy reverses the metabolic benefits in *Agrp-Lgr4*$^{-/-}$ and *Sf1-Lgr4*$^{-/-}$ mice.**

Intrascapular BAT Bilateral Sympathectomy were performed on male mice fed on HFD for 8 weeks, beginning at 6 weeks. All data were shown as mean±SEM, comparisons between more than two groups or variables were analyzed by one-way or two-way ANOVA and/or Tukey's post hoc test,$P<0.05$, * FF-sham vs *Agrp* or *Sf1-Lgr4*$^{-/-}$-sham, $^\$$ FF-sham vs FF-Surgery, $^\&$ *Agrp* or *Sf1-Lgr4*$^{-/-}$-sham vs *Agrp* or *Sf1-Lgr4*$^{-/-}$-Surgery, $^\#$ FF-Surgery vs *Agrp* or *Sf1-Lgr4*$^{-/-}$-Surgery. (A) Representative H&E staining of liver (scale bar: 500 pixel) sections from FF and *Agrp-Lgr4*$^{-/-}$ mice with or without surgery. (B) Triglyceride content (left, *$P = 0.0102$; $^\&P = 0.0197$) and total cholesterol content (right, *$P = 0.0076$) in liver, FF-sham $n = 5$, *Agrp-Lgr4*$^{-/-}$-sham $n = 5$, FF-Surgery $n = 5$, *Agrp-Lgr4*$^{-/-}$-Surgery $n = 5$. (C) Triglyceride content (left, *$P = 0.0393$) and total cholesterol content (right, *$P = 0.0093$) in plasma, FF-sham $n = 5$, *Agrp-Lgr4*$^{-/-}$-sham $n = 5$, FF-Surgery $n = 5$, *Agrp-Lgr4*$^{-/-}$-Surgery $n = 5$. (D) Expression of genes related to lipid genesis, transport and β-oxidation in liver tissue, FF-sham $n = 4$, *Agrp-Lgr4*$^{-/-}$-sham $n = 4$, FF-Surgery $n = 4$, *Agrp-Lgr4*$^{-/-}$-Surgery $n = 4$ (Gpat: *$P = 0.0437$; $^\&P = 0.0153$. Dgat1: *$P = 0.0272$; $^\&P = 0.0429$. Atgl: *$P = 0.0020$; $^\&P < 0.0001$. Cpt1α: *$P = 0.0018$; $^\&P < 0.0001$). (E) Representative H&E staining of liver (scale bar: 500 pixel) sections from FF and *Sf1-Lgr4*$^{-/-}$ mice with or without surgery. (F) Triglyceride content (left, *$P = 0.0007$; $^\&P = 0.0176$) and total cholesterol content (right, $^\&P = 0.0164$) in liver, FF-Sham $n = 6$, FF-Surgery $n = 6$, *Sf1-Lgr4*$^{-/-}$-Sham $n = 5$, *Sf1-Lgr4*$^{-/-}$-Surgery $n = 6$. (G) Triglyceride content (left, *$P = 0.05$; $^\&P = 0.0067$) and total cholesterol content (right) in plasma, FF-Sham $n = 6$, FF-Surgery $n = 6$, *Sf1-Lgr4*$^{-/-}$-Sham $n = 5$, *Sf1-Lgr4*$^{-/-}$-Surgery $n = 6$. (H) Expression of genes related to lipid genesis, transport and β-oxidation in liver tissue, FF-Sham $n = 3$, FF-Surgery $n = 3$, *Sf1-Lgr4*$^-$-Sham $n = 3$, *Sf1-Lgr4*$^{-/-}$-Surgery $n = 3$ (Dgat1: $^{\$\#}P < 0.0001$; $^\&P = 0.0037$. Atgl: $^{\$\,\&}P < 0.0001$.Cpt1α: *$P = 0.0159$). (I) Representative H&E staining of SAT (scale bar: 500 pixel) sections from FF and *Agrp-Lgr4*$^{-/-}$ mice with or without surgery. (J) Adipocyte size of SAT, FF-sham $n = 5$, *Agrp-Lgr4*$^{-/-}$-sham $n = 5$, FF-Surgery $n = 5$, *Agrp-Lgr4*$^{-/-}$-Surgery $n = 5$ (1–20 μm: *$P = 0.0007$, $^\&P = 0.0034$. 20-40 μm: *$^\&P < 0.0001$; $^\#P = 0.0019$. 40–60 μm: *$P = 0.0134$. $^{\&\,\#}P < 0.0001$; 60-80 μm: *$P < 0.0001$; $^\&P = 0.0153$; $^\#P = 0.0302$. 80–100 μm: *$P = 0.0089$; $^\&P = 0.0032$). (K) Expression of genes related to lipolysis and browning in SAT, FF-sham $n = 3$, *Agrp-Lgr4*$^{-/-}$-sham $n = 3$, FF-Surgery $n = 3$, *Agrp-Lgr4*$^{-/-}$-Surgery $n = 3$ (Adipoq: $^\&P = 0.0137$; $^\#P = 0.0024$. Adrb3: *$P = 0.0043$; $^\&P < 0.0001$; Cidea: *$P = 0.0464$; $^\&P = 0.0046$. Pgc-1α: *$P = 0.0313$; $^\&P = 0.0013$. Pgc-1β: $^\#P = 0.0039$). (L) Plasma leptin levels, FF-sham $n = 5$, *Agrp-Lgr4*$^{-/-}$-sham $n = 5$, FF-Surgery $n = 5$, *Agrp-Lgr4*$^{-/-}$-Surgery $n = 5$ (*$P = 0.0005$; $^\#P = 0.0042$). (M) Representative H&E staining of SAT (scale bar: 200 pixel) sections from FF and *Sf1-Lgr4*$^{-/-}$ mice with or without surgery. (N) Adipocyte size of SAT, FF-Sham $n = 6$, FF-Surgery $n = 6$, *Sf1-Lgr4*$^{-/-}$-Sham $n = 5$, *Sf1-Lgr4*$^{-/-}$-Surgery $n = 6$ (1–20 μm: *$P = 0.0263$; $^\&P = 0.0349$. 20–40 μm: *$P = 0.0109$; $^\&P = 0.0006$. 100–120 μm: $^\&P = 0.0108$). (O) Expression of genes related to lipolysis and browning in SAT, FF-Sham $n = 4$, FF-Surgery $n = 4$, *Sf1-Lgr4*$^{-/-}$-Sham $n = 4$, *Sf1-Lgr4*$^{-/-}$-Surgery $n = 4$ (Cidea: *$P = 0.0422$; $^\&P = 0.0086$. Pgc-1α: *$^\&P < 0.0001$. Pgc-1β: *$^\&P < 0.0001$). (P) Plasma leptin levels, FF-Sham $n = 6$, FF-Surgery $n = 6$, *Sf1-Lgr4*$^{-/-}$-Sham $n = 5$, *Sf1-Lgr4*$^{-/-}$-Surgery $n = 6$ (*$P = 0.0107$; $^\&P = 0.0028$). Source data are available online for this figure.

