## [Peer Review File · EMBO Reports]

Deficiency of neuronal LGR4 reduces obesity via hypothalamic leptin signaling

Weizhen Zhang, Li-Ping Zhang, Yuan Li, Wenbin Gao, Ziru Li, Tong Wu, Chunhui Lang, and Liangyou Rui

Corresponding author(s): Weizhen Zhang (weizhenz@med.umich.edu) , Liangyou Rui (ruily@umich.edu)

Review Timeline:

Submission Date:	4th Jun 24
Editorial Decision:	31st Jul 24
Revision Received:	11th Oct 24
Editorial Decision:	27th Nov 24
Revision Received:	18th Dec 24
Accepted:	24th Jan 25

Editor: Deniz Senyilmaz Tiebe

Transaction Report:

Dear Prof. Zhang,

Thank you for submitting your research manuscript to our journal, which was now seen by two referees, whose reports are copied below.

I apologize for this unusual delay in getting back to you, it took longer than anticipated to receive the referee reports given this busy time of the year.

Referees express interest in the proposed effects of neuronal LGR4 depletion on hypothalamic leptin signaling. However, they also raise some concerns that need to be addressed to consider publication here.

I find the reports informed and constructive, and believe that addressing the concerns raised will significantly strengthen the manuscript. As the reports are below, and I think all points need to be addressed, I will not detail them here. Please contact me if you have questions or comments regarding the revision for further discussion (also by video chat).

Given these positive recommendations, we would like to invite you to submit a revised manuscript. Please revise your manuscript with the understanding that the referee concerns (as in their reports) must be fully addressed and their suggestions taken on board. Please address all referee concerns in a complete point-by-point response. Acceptance of the manuscript will depend on a positive outcome of a second round of review. It is EMBO reports policy to allow a single round of major experimental revision only and acceptance or rejection of the manuscript will therefore depend on the completeness of your responses included in the next, final version of the manuscript.

We realize that it is difficult to revise to a specific deadline. In the interest of protecting the conceptual advance provided by the work, we recommend a revision within 3 months. Please discuss the revision progress ahead of this time with me if you require more time to complete the revisions, or if you have questions or comments regarding the revision (also by video chat).

1. A data availability section providing access to data deposited in public databases is missing (where applicable).
2. Your manuscript contains statistics and error bars based on $n=2$. Please use scatter plots in these cases.

You can submit the revision either as a Scientific Report or as a Research Article. For Scientific Reports, the revised manuscript can contain up to 5 main figures and 5 Expanded View figures, and it should not exceed 27000 characters. If the revision leads to a manuscript with more than 5 main figures it will be published as a Research Article. In this case the Results and Discussion section should be separate. If a Scientific Report is submitted, these sections have to be combined. This will help to shorten the manuscript text by eliminating some redundancy that is inevitable when discussing the same experiments twice. In either case, all materials and methods should be included in the main manuscript file.

<<https://www.embopress.org/page/journal/14693178/authorguide#expandedview>>

4) a .docx formatted letter INCLUDING the reviewers' reports and your detailed point-by-point responses to their comments. As part of the EMBO publication's Transparent Editorial Process, EMBO reports publishes online a Review Process File (RPF) to accompany accepted manuscripts. This File will be published in conjunction with your paper and will include the referee reports, your point-by-point response and all pertinent correspondence relating to the manuscript.

<https://www.embopress.org/page/journal/14693178/authorguide#transparentprocess>

5) a complete author checklist, which you can download from our author guidelines

<https://www.embopress.org/page/journal/14693178/authorguide>. Please insert information in the checklist that is also reflected in the manuscript. The completed author checklist will also be part of the RPF.

6) Please note that all corresponding authors are required to supply an ORCID ID for their name upon submission of a revised manuscript (<<https://orcid.org/>>). Please find instructions on how to link your ORCID ID to your account in our manuscript tracking system in our Author guidelines

<<https://www.embopress.org/page/journal/14693178/authorguide#authorshipguidelines>>

7) Before submitting your revision, primary datasets produced in this study need to be deposited in an appropriate public database (see <https://www.embopress.org/page/journal/14693178/authorguide#datadeposition>). Please remember to provide a reviewer password if the datasets are not yet public. The accession numbers and database should be listed in a formal "Data Availability" section placed after Materials & Method (see also

<https://www.embopress.org/page/journal/14693178/authorguide#datadeposition>). Please note that the Data Availability Section is restricted to new primary data that are part of this study. * Note - All links should resolve to a page where the data can be accessed. *

Additional information on source data and instruction on how to label the files are available:

<https://www.embopress.org/page/journal/14693178/authorguide#sourcedata>

9) Our journal encourages inclusion of *data citations in the reference list* to directly cite datasets that were re-used and obtained from public databases. Data citations in the article text are distinct from normal bibliographical citations and should directly link to the database records from which the data can be accessed. In the main text, data citations are formatted as follows: "Data ref: Smith et al, 2001" or "Data ref: NCBI Sequence Read Archive PRJNA342805, 2017". In the Reference list, data citations must be labeled with "[DATASET]". A data reference must provide the database name, accession number/identifiers and a resolvable link to the landing page from which the data can be accessed at the end of the reference. Further instructions are available at <http://www.embopress.org/page/journal/14693178/authorguide#referencesformat>

10) Regarding data quantification (see Figure Legends:

<https://www.embopress.org/page/journal/14693178/authorguide#figureformat>)

12) Please also note our reference format:

13) All Materials and Methods need to be described in the main text using our 'Structured Methods' format, which is required for all research articles. According to this format, the Methods section includes a Reagents and Tools Table (listing key reagents, experimental models, software and relevant equipment and including their sources and relevant identifiers) followed by a Methods and Protocols section describing the methods using a step-by-step protocol format. The aim is to facilitate adoption of the methodologies across labs. More information on how to adhere to this format as well as a downloadable template (.docx) for the Reagents and Tools Table can be found in our author guidelines:

I look forward to seeing a revised version of your manuscript when it is ready. Please let me know if you have questions or comments regarding the revision.

Kind regards,

Deniz Senyilmaz Tiebe

Deniz Senyilmaz Tiebe, PhD
Senior Scientific Editor
EMBO Reports

Referee #1:

This study provides significant insights into the role of leucine-rich repeat-containing G protein-coupled receptor 4 (LGR4) in regulating energy balance and metabolism. The study employs a comprehensive approach, examining the effects of LGR4 knockdown in different neuronal populations, including nestin, Sp1, AgRP, and Sf1. Altogether the data highlights the impact of LGR4 deficiency on energy expenditure, food intake, glucose metabolism, and lipid metabolism. Mechanistically, LGR4 deficiency improved leptin signaling, decreased hypothalamic SOCS-3 levels, and increased pSTAT3.

Generally, the study is viewed as significant and valuable highlighting potential novel targets for obesity and metabolic disorders. Few comments:

Given the role of LGR4 in food intake, where there any changes in feeding behavior or meal patterns?

A more comprehensive discussion on the sex specific differences in LGR4's role in metabolism would be beneficial. Why did the reduction in food intake in Sp1-Lgr4^{-/-} mice appear more pronounced in males compared to females? What potential compensatory mechanisms might be in play in female Sp1-Lgr4^{-/-} mice? Why were the effects of Lgr4 knockdown in AgRP neurons not observed in female mice?

What are the implications of the reversal of lean phenotype in Sf1-Lgr4^{-/-} mice following sympathectomy?

Are there alternative pathways or interventions that can mimic the effects of Lgr4 deficiency in Sf1 neurons without requiring sympathectomy?

What downstream signaling pathways are activated by increased leptin sensitivity in LGR4-deficient neurons?

Referee #2:

In this study, Zhang and Li et al examine the role of LGR4 in neurons by knocking it out using various neuronal models. Their results are remarkably consistent, showing that the LGR4 knockout leads to increased energy expenditure in all models, and reduced food intake in most models. This manuscript convincingly shows that central LGR4 deficiency reduces HFD-induced weight gain, but some questions remain regarding mechanism and tissue specificity. Specific comments are as follows:

1. It appears that in all the models used here, the phenotyping experiments (GTT, ITT, energy expenditure) were performed after weight divergence. While for some models, like the Nestin-Lgr4^{-/-} this is unavoidable as the weights diverge at the start of HFD.

For models like the Sf1-Lgr4^{-/-} it would be interesting to see if the differences in energy expenditure and glucose tolerance are seen before the weight divergence. Are these mice insulin tolerant because of the knockout itself or just because the knockout leads to a difference in body weight?

2. How does weight loss compare between the whole-body knockout model and the nestin knockout? In other words, do the authors think that other tissues that express LGR4 contribute to the weight loss phenotype, or do they think it is largely due to central effects?
3. Were the metabolic cage experiments performed on the Sf1-Lgr4 bilateral sympathectomy mice from figure 6? If so, were they consistent with the other data?
4. When the authors plot metabolic cage data in figures 1-5 the y axis is normalized by weight. Please clarify if this is lean weight or total weight, if this is total weight please instead normalize to lean weight. (See PMID: 20103710 for more information).
5. The beginning of the discussion says: "The metabolic benefits of global LGR4 deficiency have been proposed to occur via activation of white adipose tissue browning. However, the presence of LGR4 receptor in hypothalamic neurons suggests an alternative mechanism may exist involving neuronal circuits." I would suggest that these two findings are not as separate as the authors make them out to be. After all, they do see an increase in adipose thermogenesis in all models they look at, and this likely causes the increase in energy expenditure they see in the metabolic cages.
6. Related to the above, in the text the authors regularly refer to thermogenic genes like Ucp1 and Cidea as lipolytic genes. While there is a relationship between thermogenesis and lipolysis, these would be more accurately described as thermogenic genes. Similarly, in the discussion they say "In particular, leptin signaling in ARC and VMH increases sympathetic output and subsequent lipolysis in brown and beige adipose tissues.". Again, I would say that what the authors are seeing is adipose tissue thermogenesis. There is often a corresponding increase in lipolysis that is needed to provide the fuel for the thermogenesis.
7. The authors use the leptin experiments in figure 7 to conclude that Lgr4 knockout leads to increased leptin sensitivity in these neurons, leading to decreased food intake which results in the presented phenotypes. However, the Sf1-Lgr4^{-/-} mice do not have reduced food intake. It would not be expected for VMH neurons to reduce food intake, but how else do the authors explain the increase in energy expenditure seen in these animals?

Weizhen Zhang M.D., Ph. D
Research Professor
NCRC 026-241N, 2800 Plymouth Rd.
Department of Surgery
University of Michigan Medical Center
Ann Arbor, MI 48109

Oct 15, 2024

Dear Dr. Bernd Pulverer,

We are pleased to submit to your outstanding journal our revised manuscript entitled **“Deficiency of neuronal LGR4 increases energy expenditure and inhibits food intake via hypothalamic leptin signaling”**. We addressed the questions of the reviewers’ point by point, please find below for the detail.

Referee #1:

This study provides significant insights into the role of leucine-rich repeat-containing G protein-coupled receptor 4 (LGR4) in regulating energy balance and metabolism. The study employs a comprehensive approach, examining the effects of LGR4 knockdown in different neuronal populations, including nestin, Sp1, AgRP, and Sf1. Altogether the data highlights the impact of LGR4 deficiency on energy expenditure, food intake, glucose metabolism, and lipid metabolism. Mechanistically, LGR4 deficiency improved leptin signaling, decreased hypothalamic SOCS-3 levels, and increased pSTAT3.

Generally, the study is viewed as significant and valuable highlighting potential novel targets for obesity and metabolic disorders.

Few comments:

Given the role of LGR4 in food intake, were there any changes in feeding behavior or meal patterns?

Reply: We did not observe any difference in meal patterns as shown in the data below:

Figure 1 Knockdown of LGR4 in neurons doesn't change meal patterns in mice fed HFD

Male mice were fed on HFD, beginning at 6 weeks old for 16 weeks. Mice were placed in TSE chamber for 7 days, food intake data were recorded in the last 3 days and analyzed. All data were shown as mean±SEM, *P<0.05. **A** FF n=4, Nestin-Lgr4^{-/-} n=4, Male. **B** FF n=4, Nestin-Lgr4^{-/-} n=4, Female. **C** FF n=3, Sp1-Lgr4^{-/-} n=3, Male. **D** FF n=4, Agrp-Lgr4^{-/-} n=4, Male. **E** FF n=4, Agrp-Lgr4^{-/-} n=4, Female. **F** FF n=6, Sf1-Lgr4^{-/-} n=6, Male. **G** FF n=6, Sf1-Lgr4^{-/-} n=6, Female.

A more comprehensive discussion on the sex specific differences in LGR4's role in metabolism would be beneficial. Why did the reduction in food intake in Sp1-Lgr4^{-/-} mice appear more pronounced in males compared to females? What potential compensatory mechanisms might be in play in female Sp1-Lgr4^{-/-} mice? Why were the effects of Lgr4 knockdown in AgRP neurons not observed in female mice?

Reply: In our hand, female Sp1-Lgr4^{-/-} and AgRP-Lgr4^{-/-} mice showed gentle phenotype than male mice. This observation is consistent with other reports showing that male and female C57BL/6J mice exhibit strikingly different responses in weight, food consumption, locomotor activity, energy expenditure, etc (doi: 10.1016/j.jdiacomp.2020.107795; doi:10.1152/ajpendo.00366.2013). Mechanism underlying the distinct responses between male and female mice may be attributed to the hormonal physiological differences. A series of evidence has suggested that estrogen and G-protein coupled estrogen receptor may have a direct effect on

the metabolic physiology including food intake (doi: 10.1016/j.yhbeh.2014.02.004. doi: 10.1073/pnas.0610787104. doi: 10.1016/j.cmet.2011.08.009.). Other mechanism such as chromosomal genetic differences (doi: 10.1046/j.1440-1746.2003.03198.x. doi:10.1016/j.genm.2009.02.002. doi: 10.1111/j.1748-1716.2010.02237.x. doi: 10.1016/j.physbeh.2009.06.010.) may also contribute to the sex difference in metabolic phenotype. We have addressed these potential mechanisms and heighted these words in lines 17-27, Page 20 and lines 1-2, Page 21 in the discussion section.

What are the implications of the reversal of lean phenotype in Sf1-Lgr4^{-/-} mice following sympathectomy?

Reply: Reversal of lean phenotype in Sf1-Lgr4^{-/-} mice following sympathectomy indicates that Sf1-Lgr4^{-/-} reduces HFD induced obesity and improves energy expenditure via VMH-SNS-BAT axis. This observation is in line with the concept that VMH control thermogenesis and energy balance via the sympathetic nerves innervating BAT.

Are there alternative pathways or interventions that can mimic the effects of Lgr4 deficiency in Sf1 neurons without requiring sympathectomy?

Reply: Although intrascapular BAT bilateral sympathectomy is the method most commonly used to demonstrate the contribution of SNS-BAT axis to metabolic phenotype, alternative approach such as electrophysiological recording of the SNS nerves innervating the BAT exists.

Weizhen Zhang M.D., Ph. D
Research Professor
NCRC 026-241N, 2800 Plymouth Rd.
Department of Surgery
University of Michigan Medical Center
Ann Arbor, MI 48109

What downstream signaling pathways are activated by increased leptin sensitivity in LGR4-deficient neurons?

Reply: Our observation (Figure 8) indicates that improved leptin sensitivity in Lgr4^{-/-} mice may increase the expression of TH via SOCS-STAT3 signaling. This concept is further supported by the observation that intrascapular BAT bilateral sympathectomy reverses the metabolic benefits of LGR4 deficiency in both AgRP and Sf-1 neurons.

Referee #2:

In this study, Zhang and Li et al examine the role of LGR4 in neurons by knocking it out using various neuronal models. Their results are remarkably consistent, showing that the LGR4 knockout leads to increased energy expenditure in all models, and reduced food intake in most models. This manuscript convincingly shows that central LGR4 deficiency reduces HFD-induced weight gain, but some questions remain regarding mechanism and tissue specificity.

Specific comments are as follows:

1. It appears that in all the models used here, the phenotyping experiments (GTT, ITT, energy expenditure) were performed after weight divergence. While for some models, like the Nestin-Lgr4^{-/-} this is unavoidable as the weights diverge at the start of HFD. For models like the Sf1-Lgr4^{-/-} it would be interesting to see if the differences in energy expenditure and glucose

tolerance are seen before the weight divergence. Are these mice insulin tolerant because of the knockout itself or just because the knockout leads to a difference in body weight?

Reply: All phenotyping experiments (GTT, ITT, energy expenditure) were performed on mice fed on NCD or HFD for 16 weeks. Although we did not detect the insulin tolerance before weight divergence, our observation that Lgr4 deficient mice fed on HFD showed a more obvious phenotype indicates that the improvement in insulin tolerance may be the consequence of reduction in body weight.

2. How does weight loss compare between the whole-body knockout model and the nestin knockout? In other words, do the authors think that other tissues that express LGR4 contribute to the weight loss phenotype, or do they think it is largely due to central effects?

Reply: Weight loss in Lgr4 deficient mice in nestin neurons was less significant than the sex matched whole-body knockout mice fed on either NCD or HFD. We thus proposed that LGR4 in peripheral tissues also contributes the body weight control. Indeed, our recent study has demonstrated that deficiency of intestinal Lgr4 reduces body weight and protects mice from HFD-induced obesity (doi.org/10.1038/s41467-024-48622-5).

3. Were the metabolic cage experiments performed on the Sf1-Lgr4 bilateral sympathectomy mice from figure 6? If so, were they consistent with the other data?

Reply: We did not perform the metabolic cage experiments performed on the Sf1-Lgr4 bilateral sympathectomy mice.

Weizhen Zhang M.D., Ph. D
Research Professor
NCRC 026-241N, 2800 Plymouth Rd.
Department of Surgery
University of Michigan Medical Center
Ann Arbor, MI 48109

4. When the authors plot metabolic cage data in figures 1-5 the y axis is normalized by weight.

Please clarify if this is lean weight or total weight, if this is total weight please instead normalize to lean weight. (See PMID: 20103710 for more information).

Reply: All metabolic cage data were normalized with total body weight. Per suggestion, we now included the data normalized with lean weight (Appendix figure 6). Consistently, knockdown of Lgr4 in nestin/Sp-1/AgRP/Sf-1 neurons increased energy expenditure either normalized to total body weight or lean weight.

Figure 2. TSE phenotype data of mice with knockdown of LGR4 in estin/Sp-1/AgRP/Sf-1 specific neurons.

A-L Mice fed HFD were placed in TSE chamber for 7 days, data for the last 3 days were recorded and analyzed. All data were shown as mean±SEM, *P<0.05. **A-C**, FF n=4, Nestin-Lgr4^{-/-} n=4. **A** O₂ consumption (Upper: male; Lower: female). **B** CO₂ production (Upper: male; Lower: female). **C** Energy expenditure (Upper: male; Lower: female). **D-F**, FF n=3, Sp1-Lgr4^{-/-} n=3. **D** O₂ consumption. **E** CO₂ production. **F** Energy expenditure. **G-I**, FF n=4, Agrp-Lgr4^{-/-} n=4. **G** O₂ consumption (Upper: male; Lower: female). **H** CO₂ production (Upper: male; Lower: female). **I** Energy expenditure (Upper: male; Lower: female). **J-L**, FF n=6, Sf1-Lgr4^{-/-} n=6. **J** O₂ consumption (Upper: male; Lower: female). **K** CO₂ production (Upper: male; Lower: female). **L** Energy expenditure (Upper: male; Lower: female).

M-O Mice fed HFD with or without Intrascapular BAT Bilateral Sympathectomy were placed in TSE chamber for 7 days, data for the last 3 days were recorded and analyzed. All data were shown as mean±SEM, *P<0.05, * FF-sham vs Agrp-Lgr4^{-/-}-sham, \$ FF-sham vs FF-Surgery, & Agrp-Lgr4^{-/-}-sham vs Agrp-Lgr4^{-/-}-Surgery, # FF- Surgery vs Agrp-Lgr4^{-/-}-Surgery, n=4. **M** O₂ consumption. **N** CO₂ production. **O** Energy expenditure.

5. The beginning of the discussion says: "The metabolic benefits of global LGR4 deficiency have been proposed to occur via activation of white adipose tissue browning. However, the presence of LGR4 receptor in hypothalamic neurons suggests an alternative mechanism may exist involving neuronal circuits." I would suggest that these two findings are not as separate as the

Weizhen Zhang M.D., Ph. D
Research Professor
NCRC 026-241N, 2800 Plymouth Rd.
Department of Surgery
University of Michigan Medical Center
Ann Arbor, MI 48109

authors make them out to be. After all, they do see an increase in adipose thermogenesis in all models they look at, and this likely causes the increase in energy expenditure they see in the metabolic cages.

Reply: Per suggestion, we have revised the statement as: "However, it remains unclear whether this is the direct effect of LGR4 on adipose tissue or indirect effect from LGR4 in other tissues. The presence of LGR4 receptor in hypothalamic neurons suggests an alternative mechanism may exist involving neuronal circuits" in lines 10-13, page 18.

6. Related to the above, in the text the authors regularly refer to thermogenic genes like Ucp1 and Cidea as lipolytic genes. While there is a relationship between thermogenesis and lipolysis, these would be more accurately described as thermogenic genes. Similarly, in the discussion they say "In particular, leptin signaling in ARC and VMH increases sympathetic output and subsequent lipolysis in brown and beige adipose tissues.". Again, I would say that what the authors are seeing is adipose tissue thermogenesis. There is often a corresponding increase in lipolysis that is needed to provide the fuel for the thermogenesis.

Reply: Per suggestion, we have omitted the lipolytic genes and lipolysis in all text.

7. The authors use the leptin experiments in figure 7 to conclude that Lgr4 knockout leads to increased leptin sensitivity in these neurons, leading to decreased food intake which results in the presented phenotypes. However, the Sf1-Lgr4^{-/-} mice do not have reduced food intake. It would not be expected for VMH neurons to reduce food intake, but how else do the authors explain the increase in energy expenditure seen in these animals?

Weizhen Zhang M.D., Ph. D
Research Professor
NCRC 026-241N, 2800 Plymouth Rd.
Department of Surgery
University of Michigan Medical Center
Ann Arbor, MI 48109

Reply: Consistent with other reports showing that VMH regulates body weight and metabolism mainly by modulating the thermogenesis, our data showed that Sf1-Lgr4^{-/-} mice fed on NCD or HFD showed a higher leptin sensitivity and SNS activity to increase brown adipose thermogenesis. (The following studies detect no effect of VMH on acute food intake: doi: 10.1016/j.molmet.2014.07.004., doi: 10.1016/j.molmet.2015.09.001, doi: 10.1016/j.neuron.2005., doi: 10.1172/JCI62848., doi: 10.1038/nn.2847., doi: 10.1016/j.cmet.2011.06.014., doi: 10.1016/j.cmet.2010.05.002).

Thank you for considering our manuscript. Please do not hesitate to contact us if we may be of further assistance.

Sincerely,

Weizhen Zhang M.D., Ph.D.
Research Professor, General Surgery

Dear Prof. Zhang,

Thank you for submitting your revised manuscript. I have now read your point-by-point response carefully and I appreciate your response to the referee comments. However, I need you to address the points below as well before I can accept the manuscript.

- Please provide 3-5 keywords for your study. These will be visible in the html version of the paper and on PubMed and will help increase the discoverability of your work.
- Please rename the 'Conflicts of Interest' section as 'Disclosure Statement and Competing Interests'.
- We note an author name spelling discrepancy between the manuscript text and our manuscript tracking system - i.e. Chuihui Lang in the manuscript text vs. Chunhui Lang in the manuscript tracking system.
- Please remove the 'Author's contribution' section from the manuscript text.
- As per our format requirements, in the reference list, citations should be listed in alphabetical order and then chronologically, with the authors' surnames and initials inverted; where there are more than 10 authors on a paper, 10 will be listed, followed by 'et al.'. Please see <https://www.embopress.org/page/journal/14693178/authorguide#referencesformat>
- We note that the Author Checklist is currently incomplete - please fill in column D as well.
- We note that ORCID iD of Dr. Weizhen Zhang has not been linked to our manuscript tracking system. EMBO Press policy asks for all corresponding authors to link to their ORCID iDs. You can read about the change under "Authorship Guidelines" in the Guide to Authors here: <https://www.embopress.org/page/journal/14693178/authorguide#authorshipguidelines>

In order to link your ORCID iD to your account in our manuscript tracking system, please do the following:

1. Click the 'Modify Profile' link at the bottom of your homepage in our system.
2. On the next page you will see a box halfway down the page titled ORCID*. Below this box is red text reading 'To Register/Link to ORCID, click here'. Please follow that link: you will be taken to ORCID where you can log in to your account (or create an account if you don't have one)
3. You will then be asked to authorise Wiley to access your ORCID information. Once you have approved the linking, you will be brought back to our manuscript system.

We regret that we cannot do this linking on your behalf for security reasons.

- We note the following regarding the funding information: Grants section and heading need to be removed from the manuscript and the funding information should be moved under the Acknowledgments section; P30DK089503 funding info needs to be removed from the Comments box in the manuscript tracking system and needs to be provided as a separate funder.
- Please provide the main and EV figures individually (as one production quality file per figure) without the legends in the separate files. The legends should remain in the manuscript text. Please see <https://www.embopress.org/page/journal/14693178/authorguide#figureformat> for further information.
- We note that Expanded View figures are currently called out in the text as Expand View Figure and so on. The correct nomenclature of Expanded View figures is Fig EV1 (and so on).
- We note that Fig EV2P is currently not called out in the text.
- We note that a Supplementary Table 1 is called out, but not provided.
- We note the following regarding the Appendix file: the PDF with Appendix Figures and legends should have a Table of Contents with page numbers and the nomenclature. The in-text callouts of the Appendix figures need to be corrected: the "S" is missing, it should be Appendix Figure S1, etc. instead of Appendix Figure 1, etc.
- The Reagents & Tools table needs to be removed from the manuscript text and uploaded separately.
- We note the following regarding the source data: The numerical data are currently in a single excel file, it should be divided into one file per figure and included in the corresponding source data folder. Moreover, the blots submitted as source data seem to be cropped almost to the same extent of the figure panels whereas the uncropped versions were provided in a single PDF. The uncropped versions should be included in the corresponding figure folders as well.
- Figure legends need to be placed at the end of the manuscript, after the References.
- We note a potential re-use of the -actin blot between Figure 4A and Figure 7L. However, this is not the case in the provided source data file (the uncropped blot PDF). Please double check and clarify.

- Our production/data editors have asked you to clarify several points in the figure legends:
 - o Please note that information related to n is missing in the legends of figures EV 2I-n.
 - o Please note that the scale bar is missing for figure 1p.
 - o Please note that scale bar and its definition are missing for figures 2o; 3o; 4o; 5m; 6h; EV 2a, d; EV 3a, e; EV 4a, e, i, m.
- Please remove the synopsis image and the text from the manuscript text.

- Please provide an image for the synopsis separately. This image should provide a rapid overview of the question addressed in the study but still needs to be kept fairly modest since the image size cannot exceed 550 (width) x 300-600 (height) pixels.

Thank you again for giving us to consider your manuscript for EMBO Reports, I look forward to your minor revision.

Kind regards,

Deniz Senyilmaz Tiebe

--

Deniz Senyilmaz Tiebe, PhD
Senior Scientific Editor
EMBO Reports

All editorial and formatting issues were resolved by the authors.

Prof. Weizhen Zhang
Department of Surgery
University of Michigan Medical Center
NCRC 026-241N, 2800 Plymouth Rd.
Ann Arbor, MI 48109
United States

Dear Prof. Zhang,

Thank you for submitting your revised manuscript. I have now looked at everything and all is fine. Therefore, I am very pleased to accept your manuscript for publication in EMBO Reports.

Congratulations on a nice work!

Kind regards,

Deniz Senyilmaz Tiebe

--

Deniz Senyilmaz Tiebe, PhD
Senior Scientific Editor
EMBO Reports
